

# Temporary and net sinks of atmospheric $CO_2$ due to chemical weathering in subtropical catchment with mixing carbonate and silicate lithology

Yingjie Cao[a,c,d], Yingxue Xuan[b], Changyuan Tang[a,b,c]*, Shuai Guan[e], Yisheng Peng[a,c]

[a]School of Environmental Science and Engineering, Sun Yat-Sen University, Guangzhou, China

[b]School of Geography and Planning, Sun Yat-Sen University, Guangzhou, China

[c]Guangdong Provincial Key Laboratory of Environmental Pollution Control and Remediation

Technology, Sun Yat-Sen University, Guangzhou, China

[d]Southern Marine Science and Engineering Guangdong Laboratory, Zhuhai, China

[e]Guangdong Research Institute of Water Resource and Hydropower, Guangzhou, China

**Abstract:** The study provides the major ion chemistry, chemical weathering rates and temporary and net $CO_2$ sinks in the Beijiang River, which was characterized as hyperactive region with high chemical weathering rates, carbonate and silicate mixing lithology and abundant sulfuric acid chemical weathering agent with acid deposition and acid mining drainage (AMD) origins. The total chemical weathering rate of 85.46 t·km$^{-2}$·a$^{-1}$ was comparable to other rivers in the hyperactive zones between the latitude 0-30°. Carbonate weathering rates of 61.15 t·km$^{-2}$·a$^{-1}$ contributed to about 70% of the total. The lithology, runoff and geomorphology had significant influence on the chemical weathering rate. The proportion of carbonate outcrops had significant positive correlation with the chemical weathering rate. Due to the interaction between dilution and compensation effect, significant positive linear relationship was detected between runoff and total, carbonate and silicate weathering rates. The geomorphology factors such as catchment area, average slope and hypsometric integral value (HI) had non-linear correlation on chemical weathering rate and showed





significant scale effect, which revealed the complexity in chemical weathering processes. DIC-
apportionment showed that CCW (Carbonate weathering by $CO_2$) was the dominant origin of DIC
(35%-87%) and that SCW (Carbonate weathering by $H_2SO_4$) (3%-15%) and CSW (Silicate
weathering by $CO_2$) (7%-59%) were non-negligible processes. The temporary $CO_2$ sink was 823.41
$10^3$ mol km$^{-2}$ a$^{-1}$. Compared with the "temporary" sink, the net sink of $CO_2$ for the Beijiang River
was approximately $23.18 \times 10^3$ mol km$^{-2}$ a$^{-1}$ of $CO_2$ and was about 2.82% of the "temporary" $CO_2$
sink. Human activities (sulfur acid deposition and AMD) dramatically decreased the $CO_2$ net sink
and even make chemical weathering a $CO_2$ source to the atmosphere.
**Keywords**:Chemical weathering, DIC-apportionment, $CO_2$ temporary sink, $CO_2$ net sink
**1    Introduction**

About half of the global $CO_2$ sequestration due to chemical weathering occurs in warm and

high runoff regions (Ludwig et al., 1998), so called the hyperactive regions and hotspots (Meybeck
et al., 2006). Chemical weathering driven by weak carbonic acid ($H_2CO_3$) that originates from
atmosphere $CO_2$ or soil respiration under natural conditions is a fundamental geochemical process
regulating the atmosphere-land-ocean fluxes and earth's climate (Guo et al., 2015). Carbonate and
silicate weathering define the two typical categories of chemical weathering. A profound case in
point is that from the view of the global carbon cycle, the $CO_2$ consumption due to carbonate
weathering is recognized the "temporary" sink because that the flux of $CO_2$ consumed by carbonate
dissolution on the continents is balanced by the flux of $CO_2$ released into the atmosphere from the
oceans by carbonate precipitation on the geological time scale (Cao et al., 2015; Garrels, 1983).
While the consumption of $CO_2$ during the chemical weathering of silicate rocks has been regard as



the net sink of $CO_2$ and regulates the global carbon cycle (Hartmann et al., 2009; Hartmann et al.,
2014b; Kempe and Degens, 1985; Lenton and Britton, 2006). Thus in carbonate-silicate mixing
catchment, it is essential to distinguish proportions of the two most important lithological groups,
i.e., carbonates and silicates, and evaluate the net $CO_2$ sink due to chemical weathering of silicate
(Hartmann et al., 2009).
In addition to the chemical weathering induced by $H_2CO_3$, sulfuric acid ($H_2SO_4$) of
anthropogenic origins produced by sulfide oxidation such as acid deposition caused by fossil fuel
burning and acid mining discharge (AMD) also becomes an important chemical weathering agent
in the catchment scale. Many studies have shown the importance of sulfide oxidation and subsequent
dissolution of other minerals by the resulting sulfuric acid at catchment scale (Hercod et al., 1998;
Spence and Telmer, 2005). Because depending on the fate of sulfate in the oceans, sulfide oxidation
coupled with carbonate dissolution could facilitate a release of $CO_2$ to the atmosphere (Spence and
Telmer, 2005), the carbonate weathering by $H_2SO_4$ (sulfide oxidation) plays a very important role
in quantifying and validating the ultimate $CO_2$ consumption rate. Thus, under the influence of
human activities, the combination of silicate weathering by $H_2CO_3$ and carbonate weathering by
$H_2SO_4$ controlled the net sink of atmospheric $CO_2$.
Numerous studies on chemical weathering of larger rivers have been carried out to examine
hydrochemical characteristics, chemical erosion and $CO_2$ consumption rates, and long-term climatic
evolution of the Earth in various large rivers, such as the Changjiang River (Chen et al., 2002a; Ran
et al., 2010), the Huanghe River (Zhang et al., 1995), the Pearl River (Gao et al., 2009; Xu and Liu,
2010; Zhang et al., 2007a), the Huai River (Zhang et al., 2011), the rivers of the Qinghai-Tibet
Plateau (Jiang et al., 2018; Li et al., 2011; Wu et al., 2008), the Mekong River (Li et al., 2014), the



rivers of the Alpine region (Donnini et al., 2016), the Sorocaba River (Fernandes et al., 2016), the
rivers of Baltic Sea catchment (Sun et al., 2017), the Amazon River (Gibbs, 1972; Mortatti and
Probst, 2003; Stallard and Edmond, 1981; Stallard and Edmond, 1983; Stallard and Edmond, 1987),
the Lena River (Huh and Edmond, 1999) and the Orinoco River (Mora et al., 2010). For simplicity
of calculation procedure, most of the researches have ignore the sulfuric acid induced chemical
weathering and resulted in an overestimation of $CO_2$ sink. To overcome this shortcoming of
traditional mass-balance method, we applied a DIC source apportionment procedure to discriminate
the contribution of sulfuric acid induced chemical weathering to validate the temporary and net sink
of $CO_2$ in a typical hyperactive region with carbonate-silicate mixing lithology to give a further
understanding of basin scale chemical weathering estimation.

The Pearl River located in the subtropical area in South China includes three principal rivers:

the Xijiang, Beijiang, and Dongjiang Rivers. The warm and wet climatic conditions make the pearl
river a hyperactive region in China. The three river basins have distinct geological conditions. The
Xijiang River is characterized as the carbonate-dominated area and the Dongjiang River has silicate
as the main rock type. While the Beijiang River, which is the second largest tributary of the Pearl
River, is characterized as a typical carbonate-silicate mixing basin. In addition, as the serve acid
deposition (Larssen et al., 2006) and active mining area (Li et al., 2019), chemical weathering
induced by sulfuric acid make the temporary and net sink of atmospheric $CO_2$ to be reevaluated. So
that, in this study, the Beijiang River as the hyperactive region in Southeast China with a typical
subtropical monsoon climate and carbonate-silicate mixing geologic settings was selected as the
study area. Three main objectives are summarized as follows: (1) reveal spatial-temporal variations
of major element chemistry of the river water, (2) calculate the chemical weathering rate and unravel





the controlling factors on chemical weathering processes, and (3) determinate the temporary sink of
$CO_2$ and evaluate the influence of sulfide oxidation on net sink of $CO_2$ by DIC apportionment
procedure.
**2    Study area**

The Beijiang River Basin, which is the second largest tributary of the Pearl River Basin, is

located in the southeast of China (Fig. 1). It covers a basin of 52 068 $km^2$ and has a total length of
573 km. The river basin is located in subtropical monsoon climate zone, with the mean annual
temperature across the drainage basin ranging from 14°C to 22°C, the mean annual precipitation
ranging from 1390 mm to 2475 mm. The average annual runoff is 51 billion $m^3$, with 70%-80% of
the flux occurring from April to September. This can be attributed to the fact that more than 70% of
the annual precipitation (about 1800 mm $year^{-1}$) is concentrated in the wet season (April to
September).

Lithology in the river basin are composed of limestone, sandstone, gneiss and glutenite. In the

upper basin, carbonate rock (mainly of limestone) outcrops in the west and center, while sandstone
of Devonian era and mudstone of Paleogene era outcrop in the east of upper stream. In the middle
of basin, limestone and sandstone cover most of the area, and Cretaceous volcanic rocks are found
in the tributary (Lianjiang River), mainly granite. In the lower basin, Achaen metamorphic rocks
outcrop in the west, and are composed of gneiss and schist, sandstone covers rest of area of the
lower basin. Quaternary sediments scatter along the main stream of the river. The carbonate and
silicate rock outcrops in the Beijiang River Basin was 10737 $km^2$ (28%) and 24687 $km^2$ (65%),
respectively.





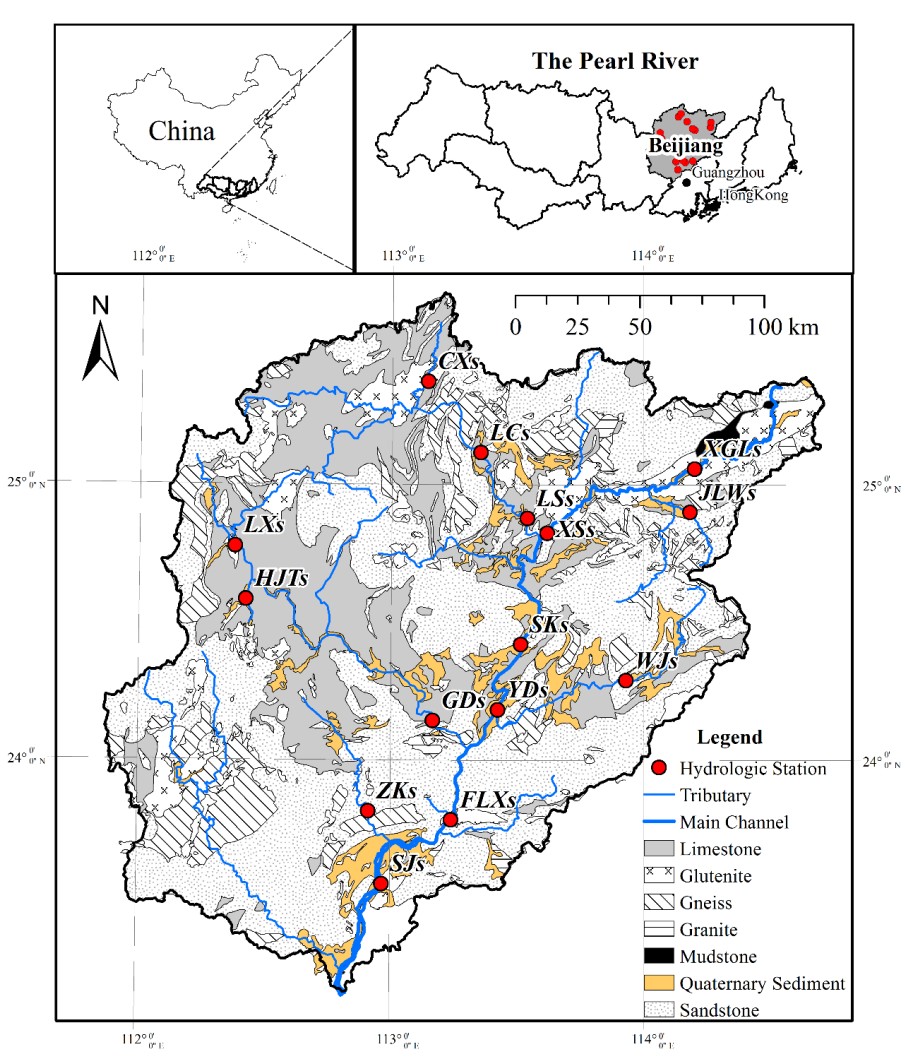

**Fig. 1 Geology map and sampling point in the Beijiang River basin by ArcGis**

## 3    Materials and methods

### 3.1    Sampling procedure and laboratory analysis

Water samples were collected monthly at 15 hydrologic stations from January to December in

2015 (Fig. 1). The river waters were sampled by a portable organic class water sampler along the

middle thread of channel in the first day of each month. In addition, to discriminate the contribution





of rain inputs, the daily rainwater was also sampled in five stations (SJs, FLXs, YDs, XSs and XGLs)
along the main stream. The rainwater collector is consisted of a funnel with diameter of 20 cm and
a 5 L plastic bottle. A rubber ball is setup in the funnel to prevent evaporation. All the river and rain
water were filtered through 0.45 μm glass fiber filter and stored in 100 ml tubes and stored below
4°C until analysis.
Electric conductivity (EC), pH and temperature (T) were measured by a multi-parameter water
quality meter (HACH-HQ40Q), and alkalinity ($HCO_3^-$) was measured in filtered water samples by
titration in situ. The cations ($Na^+$, $K^+$, $Ca^{2+}$, $Mg^{2+}$) and anions ($Cl^-$, $SO_4^{2-}$) were analyzed by ion
chromatography (ThermoFisher ICS-900) with limit of detection (L.O.D) of 0.01 mg/L. The
dissolved $SiO_2$ was measured by molybdenum yellow method and was analyzed by ultraviolet
spectrophotometer (Shimadzu UV-2600).
**3.2  Calculation procedure**
**3.2.1      Chemical weathering rates**
According to the principle of the mass balance, the mass balance equation for element $X$ in the
dissolved load can be expressed as (Galy and France-Lanord, 1999):
$$[X]_{riv} = [X]_{pre} + [X]_{eva} + [X]_{sil} + [X]_{car} + [X]_{anth} \tag{1}$$
Where [$X$] denotes the elements of $Ca^{2+}$, $Mg^{2+}$, $Na^+$, $K^+$, $Cl^-$, $SO_4^{2-}$, $HCO_3^-$ in $mmol·L^{-1}$. The
subscripts riv, pre, eva, sil, car and anth denotes the river, precipitation source, evaporite source,
silicate source, carbonate source and anthropogenic source.
On the basis of the theory of rock chemical weathering and ignoring the anthropogenic origins
of major ions (except for $SO_4^{2-}$) due to relative low TDS of river water samples (Cao et al., 2016a;





Gaillardet et al., 1999b) ranged from 73.79 to 230.16 mg·L$^{-1}$, the major elements of river water can
be simplified as followed:
$$[Cl^-]_{riv} = [Cl^-]_{pre} + [Cl^-]_{eva} \tag{2}$$
$$[K^+]_{riv} = [K^+]_{pre} + [K^+]_{sil} \tag{3}$$
$$[Na^+]_{riv} = [Na^+]_{pre} + [Na^+]_{eva} + [Na^+]_{sil} \tag{4}$$
$$[Ca^{2+}]_{riv} = [Ca^{2+}]_{pre} + [Ca^{2+}]_{sil} + [Ca^{2+}]_{car} \tag{5}$$
$$[Mg^{2+}]_{riv} = [Mg^{2+}]_{pre} + [Mg^{2+}]_{sil} + [Mg^{2+}]_{car} \tag{6}$$
$$[HCO_3^-]_{sil} = [K^+]_{sil} + [Na^+]_{sil} + 2[Mg^{2+}]_{sil} + 2[Ca^{2+}]_{sil} \tag{7}$$
$$[HCO_3^-]_{car} = [HCO_3^-]_{riv} - [HCO_3^-]_{sil} \tag{8}$$
$$[SO_4^{2-}]_{riv} = [SO_4^{2-}]_{pre} + [SO_4^{2-}]_{anth} \tag{9}$$
Firstly, the measured ion concentrations of the rain water are rectified by evaporation
coefficient α=0.63=P/R (with P the precipitation and R the runoff) and calculated the contributions
of atmospheric precipitation. Secondly, due to low TDS, the anthropogenic contributions are
negligible. Thirdly, the molar ratios of Ca$^{2+}$/Na$^+$ (0.4) and Mg$^{2+}$/Na$^+$ (0.2) for silicate end-member
(Zhang et al., 2007b) are used to calculate the contribution of Ca$^{2+}$ and Mg$^{2+}$ from silicate
weathering, and then, residual Ca$^{2+}$ and Mg$^{2+}$ were attributed to carbonate weathering. For monthly
data, the contributions of different sources can be calculated as followed:
$$R_{car}=([Ca^{2+}]_{car}+[Mg^{2+}]_{car})/S \times 100\% \tag{10}$$
$$R_{sil}=([K^+]_{sil}+[Na^+]_{sil}+[Ca^{2+}]_{sil}+[Mg^{2+}]_{sil})/S \times 100\% \tag{11}$$
$$R_{eva}=[Na^+]_{eva}/S \times 100\% \tag{12}$$
$$R_{pre}=([K^+]_{pre}+[Na^+]_{pre}+[Ca^{2+}]_{pre}+[Mg^{2+}]_{pre})/S \times 100\% \tag{13}$$
$$S = [Ca^{2+}]_{car} + [Mg^{2+}]_{car} + [Ca^{2+}]_{sil} + [Mg^{2+}]_{sil} + [Na^+]_{sil} + [K^+]_{sil} + [Na^+]_{eva} +$$



$[Ca^{2+}]_{pre} + [Mg^{2+}]_{pre} + [Na^+]_{pre} + [K^+]_{pre}$             (14)
Where $R$ denotes the proportions of dissolved cations from different sources. $S$ denotes the total
concentrations of cations for river water in mmol·L$^{-1}$.
The total, carbonate and silicate chemical weathering rates (TWR, CWR and SWR) can be
estimated as followed:
$CWR=\sum_{i=1}^{n=12}[(24 \times [Mg^{2+}]_{car}+40 \times [Ca^{2+}]_{car}+61 \times [HCO_3^-]_{car} \times 0.5)_i \times Q_i/(10^6A)]$    (15)
$SWR=\sum_{i=1}^{n=12}[(39 \times [K^+]_{sil}+23 \times [Na^+]_{sil}+24 \times [Mg^{2+}]_{sil}+40 \times [Ca^{2+}]_{sil}+96 \times [SiO_2]_{sil})_i \times$
$Q_i/(10^6A)]$                                  (16)
$TWR=CWR+SWR$                      (17)
Where TWR, CWR and SWR have the unit of t km$^{-2}$ a$^{-1}$, $Q_i$ denotes discharge in m$^3$·month$^{-1}$, and A
denotes the catchment area in km$^2$.
**3.2.2     DIC apportionments**
The riverine DIC originates from several sources including carbonate minerals, respired soil
$CO_2$ and atmospheric $CO_2$, and it could be affected by processes occurring along the water pathways
(Khadka et al., 2014; Li et al., 2008). Four dominant weathering processes, including (1) carbonate
weathering by carbonic acid (CCW), (2) carbonate weathering by sulfuric acid (SCW), (3) silicate
weathering by carbonic acid (CSW), (4) and silicate weathering by sulfuric acid (SSW), can be
described by the following reaction equations:
$CCW:(Ca_{2-x}Mg_x)(CO_3)_2 + 2H_2CO_3 \rightarrow (2-x)Ca^{2+} + xMg^{2+} + 4HCO_3^-$     (18)
$SCW:(Ca_{2-x}Mg_x)(CO_3)_2 + H_2SO_4 \rightarrow (2-x)Ca^{2+} + xMg^{2+} + 2HCO_3^- + SO_4^{2-}$   (19)
$CSW:CaSiO_3 + 2H_2CO_3 + H_2O \rightarrow Ca^{2+} + H_4SiO_4 + 2HCO_3^-$       (20)





$\quad SSW:CaSiO_3 + H_2SO_4 + H_2O \rightarrow Ca^{2+} + H_4SiO_4 + SO_4^{2-}$ (21)
Where CaSiO₃ represents an arbitrary silicate.
According to the study of (Galy and France-Lanord, 1999) and (Spence and Telmer, 2005),
carbonate and silicate weathering by carbonic acid in the same ratio as carbonate and silicate
weathering by sulfuric acid, for monthly data the mass balance equations are followed:
$\quad [SO_4^{2-}]_{riv} - [SO_4^{2-}]_{pre} = [SO_4^{2-}]_{SCW} + [SO_4^{2-}]_{SSW}$ (22)
$\quad [SO_4^{2-}]_{riv} - [SO_4^{2-}]_{pre} = \alpha_{SCW} \times [HCO_3^-]_{riv} \times 0.5 + \frac{\alpha_{CSW} \times \alpha_{SCW}}{\alpha_{CCW}} \times [HCO_3^-]_{riv}$ (23)
Where the subscripts CCW, SCW, CSW and SSW denotes the four end-members defined by
carbonate weathering by carbonic acid, carbonate weathering by sulfuric acid, silicate weathering
by carbonic acid and silicate weathering by sulfuric acid, respectively. The parameter α denotes the
proportion of DIC derived from each end-member processes.
According to the above description, the ion balance equations are followed:
$\quad [Ca^{2+}]_{car} + [Mg^{2+}]_{car} = \alpha_{CCW} \times [HCO_3^-]_{riv} \times 0.5 + \alpha_{SCW} \times [HCO_3^-]_{riv}$ (23)
$\quad [SO_4^{2-}]_{SCW} + [SO_4^{2-}]_{SSW} = \alpha_{SCW} \times [HCO_3^-]_{riv} \times 0.5 + \frac{\alpha_{CSW} \times \alpha_{SCW}}{\alpha_{CCW}} \times [HCO_3^-]_{riv}$ (24)
$\quad \alpha_{CCW} + \alpha_{SCW} + \alpha_{CSW} = 1$ (25)
Combing the above equations, the proportions of HCO₃⁻ derived from three end-members
(CCW, SCW and CSW) can be calculated, and the DIC (equivalent to HCO₃⁻) fluxes by different
chemical weathering processes are calculated by following equations.
$\quad DIC_{CCW} = \alpha_{CCW} \times [HCO_3^-]_{riv}$ (26)
$\quad DIC_{SCW} = \alpha_{SCW} \times [HCO_3^-]_{riv}$ (27)
$\quad DIC_{CSW} = \alpha_{CSW} \times [HCO_3^-]_{riv}$ (28)



### 3.2.3    CO₂ consumption rate and CO₂ net sink

According to the equations (17)~ (20), only the processes of CCW and CSW can consume

the $CO_2$ from atmosphere or soil and only half of the $HCO_3^-$ in the water due to carbonate weathering

by carbonic acid come from atmospheric $CO_2$. Thus, the $CO_2$ consumption rates (CCR) for CCW

and CSW can be calculated as followed (Zeng et al., 2016):

$$CCR_{CCW} = \sum_{i=1}^{n=12}\{[0.5 \times (Q/A) \times [HCO_3^-]_{CCW}]/1000\}_i \qquad (29)$$

$$CCR_{CSW} = \sum_{i=1}^{n=12}\{[(Q/A) \times [HCO_3^-]_{CSW}]/1000\}_i \qquad (30)$$

Where Q is discharge in $m^3 \cdot a^{-1}$, $[HCO_3^-]$ is concentration of $HCO_3^-$ in $mmol \cdot L^{-1}$, A is catchment area

in $km^2$. So that the CCR has the unit of $10^3\ mol\ km^{-2} \cdot a^{-1}$.

According to the classical view of the global carbon cycling (Berner and Kothavala, 2001),

the CCW is not a mechanism that can participate to the amount of $CO_2$ in the atmosphere because

all of the atmospheric fixed through CCW is returned to the atmosphere during carbonate

precipitation in the ocean. However, when sulfuric acid is involved as a proton donor in carbonate

weathering, half of the dissolved carbon re-release to the atmospheric during carbonate precipitation.

Thus, SCW leads to a net release of $CO_2$ in ocean-atmosphere system over timescale typical of

residence time of $HCO_3^-$ in the ocean ($10^5$ years). Meanwhile, in case of CSW, followed by

carbonate deposition, one of the two moles of $CO_2$ involved is transferred from the atmosphere to

the lithosphere in the form of carbonate rocks, while the other one returns to the atmosphere,

resulting a net sink of $CO_2$. Therefore, the net $CO_2$ consumption rate ($CCR_{Net}$) due to chemical

weathering can be concluded as followed:

$$CCR_{Net} = \sum_{i=1}^{n=12}\{[(0.5 \times [HCO_3^-]_{CSW} - 0.5 \times [HCO_3^-]_{SCW}) \times (Q/A)]/1000\}_i \qquad (31)$$





**3.3 Spatial and statistical analysis**
The hypsometric integral value (HI) (PIKE and WILSON, 1971) was employed in this study
to evaluate the influence of terrain on the chemical weathering. HI is an important index to reveal
the relationship between morphology and development of landforms and can be used to establish
the quantitative relationship between the stage of geomorphological development and the material
migration in the basin (PIKE and WILSON, 1971; Singh et al., 2008; STRAHLER, 1952). The HI
value of each watershed is calculated by the elevation-relief ratio method and can be obtained by
the following equation (PIKE and WILSON, 1971):
$$\text{HI} = \frac{\text{Mean. elevation} - \text{Min. elevation}}{\text{Max. elevation} - \text{Min. elevation}}$$
where HI is the hypsometric integral; Mean.elevation is the mean elevation of the watershed;
Min.elevation is the minimum elevation within the watershed; Max.elevation is the maximum
elevation within the watershed. According to the hypsometric integral value (HI), the
geomorphological development can be divided into three stages: inequilibrium or young stage (HI >
0.6), equilibrium or mature stage (0.35 < HI $\leqslant$ 0.6), and monadnock or old age ( HI $\leqslant$ 0.35),
which can reflect the erodible degree and erosion trend of the geomorphology (Xiong et al., 2014).
The watershed of the study area was divided by using hydrological analysis module of ArcGIS.
The average slope and HI was conducted by spatial analysis module of ArcGIS. All statistical tests
were conducted using SPSS version 22.0. One-way analysis of variance (ANOVA) was performed
for differences of monthly major ion concentrations and dissolved inorganic carbonate isotopes with
significance at p<0.05. Principal component analysis (PCA) was employed to unravel the
underlying data set through the reduced new variables, analyzed the significant factors affecting the
characteristics of water chemistry.



**4    Results**
**4.1   Chemical composition in the Beijiang River Basin**

The major physical-chemical parameters of river water samples were presented in Table 1. For

all the monthly samples, the pH values ranged from 7.5 to 8.5 with an average of 8.05. Average
electrical conductivity was 213 μs·cm⁻¹, ranging from 81 to 330 μs·cm⁻¹. The TDS of river water
samples varied from 73.8 to 230.2 mg·L⁻¹, with an average of 157.3 mg·L⁻¹, which was comparable
with the global average of 100 mg·L⁻¹ (Gaillardet et al., 1999a). Compared with the major rivers in
China, the average TDS was significantly lower than the Changjiang (Chen et al., 2002b) , the
Huanghe (He Jiangyi, 2017) the Zhujiang (Zhang et al., 2007b), the Huaihe (Zhang et al., 2011) and
the Liaohe (Ding et al., 2017). However, the average TDS was higher than the rivers draining
silicate-rock-dominated areas, e.g., Dojiang River (59.9 mg·L⁻¹) in Southern China (Xie chenji,
2013), North Han River (75.5 mg·L⁻¹) in South Korea, (Ryu et al., 2008), the Amazon (41 mg·L⁻¹)
and the Orinoco (82 mg·L⁻¹) draining the Andes (Dosseto et al., 2006; Edmond et al., 1996).






**Table 1** The major physical-chemical parameters of river water samples at 15 hydrological station in the Beijiang River (mean±SD). The total dissolved solid (TDS, mg·L⁻¹) expressed as the sum of major inorganic species concentration ($Na^+ + K^+ + Ca^{2+} + Mg^{2+} + HCO_3^- + Cl^- + SO_4^{2-} + NO_3^- + SiO_2$)

| Hydrological stations | pH | EC (µs/cm) | TDS (mg/L) | $Na^+$ (µmol/L) | $K^+$ (µmol/L) | $Ca^{2+}$ (µmol/L) | $Mg^{2+}$ (µmol/L) | $HCO_3^-$ (µmol/L) | $Cl^-$ (µmol/L) | $SO_4^{2-}$ (µmol/L) | $SiO_2$ (µmol/L) | HI |
|---|---|---|---|---|---|---|---|---|---|---|---|---|
| JLWs | 7.9±0.15 | 95±39.98 | 81.1±25.6 | 111.4 | 51.9 | 223.5 | 103.9 | 701.9 | 28.3 | 44.5 | 225.2 | 0.3444 |
| CXs | 8.2±0.15 | 219±50.32 | 163.7±20.9 | 118.1 | 40.1 | 793.3 | 187.1 | 1593.6 | 60.5 | 199.4 | 106.3 | 0.2865 |
| HJTs | 8.1±0.19 | 203±34.39 | 151.8±21.9 | 100.2 | 29.9 | 686.7 | 203.9 | 1708.7 | 29.5 | 72.2 | 156.6 | 0.2991 |
| ZKs | 8.1±0.13 | 218±44.84 | 161.3±21.1 | 426.4 | 66.2 | 560.3 | 134.1 | 1276.9 | 134.7 | 161.4 | 151.9 | 0.2233 |
| XGLs | 7.8±0.15 | 168±15.83 | 117.9±8.9 | 315.4 | 112.4 | 422.4 | 101.0 | 992.2 | 213.9 | 112.6 | 178.9 | 0.1821 |
| WJs | 8.1±0.1 | 260±26.91 | 172.9±16.7 | 197.8 | 59.0 | 767.3 | 122.6 | 1467.1 | 99.1 | 162.8 | 183.4 | 0.2462 |
| LXs | 8.1±0.16 | 236±32.99 | 171.8±19.6 | 122.1 | 38.1 | 813.5 | 176.0 | 1829.4 | 51.5 | 89.2 | 145.7 | 0.2149 |
| LCs | 8.2±0.09 | 253±25.91 | 196.1±20.0 | 287.4 | 46.8 | 862.6 | 234.4 | 1845.7 | 115.7 | 232.4 | 130.7 | 0.2731 |
| LSs | 8.3±0.06 | 220±45.62 | 184.2±18.3 | 258.9 | 58.2 | 793.5 | 202.9 | 1740.6 | 109.0 | 191.9 | 121.4 | 0.2503 |
| XSs | 7.9±0.14 | 156±29.8 | 123.9±17.6 | 305.0 | 86.1 | 366.8 | 110.9 | 966.6 | 103.8 | 166.5 | 218.7 | 0.2365 |
| GDs | 8.1±0.05 | 232±10.67 | 169.4±8.3 | 112.6 | 40.5 | 781.6 | 172.1 | 1798.5 | 44.0 | 90.3 | 141.2 | 0.2415 |
| SKs | 8.1±0.19 | 238±21.6 | 161.1±17.4 | 345.3 | 73.6 | 641.0 | 162.5 | 1304.1 | 174.4 | 223.5 | 160.1 | 0.2061 |
| Yds | 7.8±0.2 | 241±54.39 | 165.9±34.0 | 296.4 | 59.3 | 674.8 | 160.9 | 1515.0 | 118.7 | 175.9 | 144.4 | 0.2055 |
| FLXs | 8±0.21 | 232±36.99 | 161.4±22.8 | 187.6 | 95.1 | 577.0 | 135.0 | 1262.4 | 111.9 | 159.6 | 169.5 | 0.2065 |
| SJs | 8.1±0.1 | 230±26.94 | 176.4±18.9 | 355.0 | 83.4 | 663.5 | 156.2 | 1367.7 | 182.4 | 190.5 | 180.5 | 0.2057 |




Major ion composition were shown in the anion and cation ternary diagrams (Fig. 2). $Ca^{2+}$ was
the dominant cation with concentration ranging from 199 to 1107 $\mu mol \cdot L^{-1}$, accounting for
approximately 49% to 81%, with an average of 66% (in $\mu Eq$) of the total cation composition in the
river water samples. $HCO_3^-$ was the dominant anion, with concentration ranging from 640 to 2289
$\mu mol \cdot L^{-1}$. On average, it comprised 77% (59%~92%) of total anions, followed by $SO_4^{2-}$ (16%) and
$Cl^-$ (6%). The major ionic composition indicated that the water chemistry of the Beijiang River
Basin was controlled by both carbonate and silicate weathering.

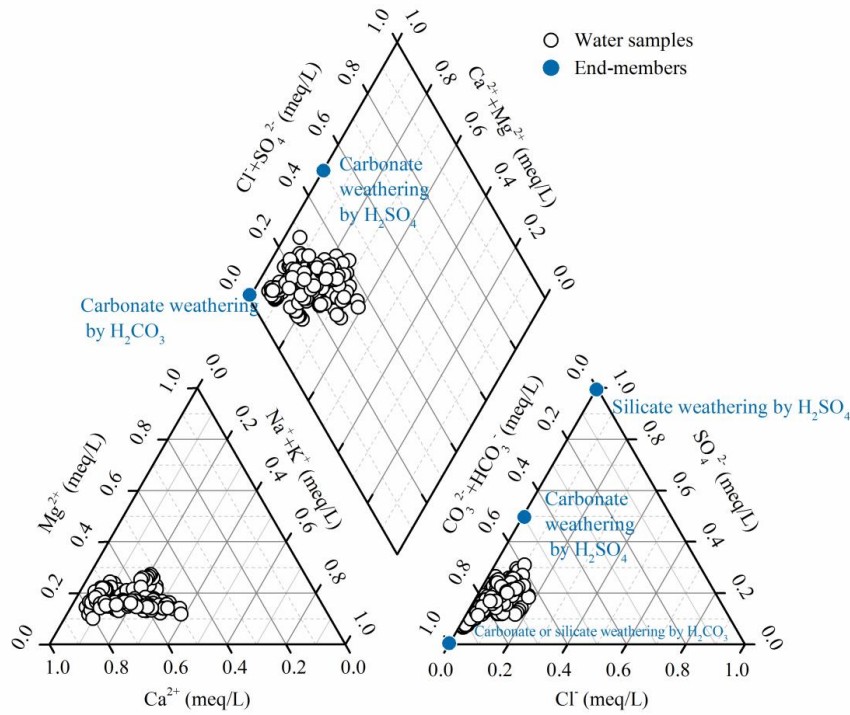


**Fig. 2 Piper diagram of river water samples in the Beijiang River**

The principal component analysis (PCA) was used to study the factors controlling the chemical
compositions. The varimax rotation was used to reduced the number of variables to two principal
components (PCs), which together explain 76.88% of the total variance in the data. The first PC



(PC1) explained approximately 50.02% of the total variations, and was considered to represent
"carbonate weathering factor" because of the high contributions of EC, TDS, $Ca^{2+}$, $Mg^{2+}$ and $HCO_3^-$
concentrations. The second PC (PC2) explained 26.85% of the total variance and presented high
loadings for $Na^+$ and $K^+$ concentrations. Thus, the PC2 represented an "silicate weathering factor",
which were considered to be two important sources of these ions in the Beijiang River Basin (Li et
al., 2019).
**4.2  Seasonal and spatial variations**
There were significantly seasonal variations in the major ion concentrations (Fig. 3). Two basic
patterns of temporal variations could be observed. The first one was related to the carbonate
weathering derived ions such as $Ca^{2+}$ and $HCO_3^-$, which showed high values in Nov and low values
in Jun. The second one was for the silicate weathering derived ions such as $Na^+$ and $K^+$, which
showed high values in Feb and low values in Jun. The minimums occurred in Jun for all the ions
showed a significant dilution effect during the high-flow periods.

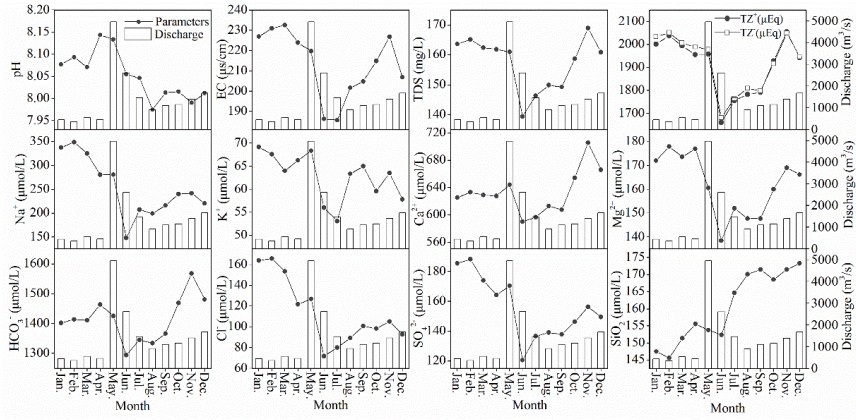


**Fig. 3 Monthly variations of environmental parameters and major ion concentrations in the**

**Beijiang River Basin (SJs station). The columns denoted the monthly discharge**



It is clear that the $Ca^{2+}$ and $HCO_3^-$ concentrations had a decreasing trend from upstream to
downstream (Fig. 4), this characteristic agrees with the trends observed in the Changjiang River and
the Huai River, where the major elements or TDS concentrations of the main channel showed a
general decreasing trend, and the tributaries display the dilution effect to the main channel. For other
silicate weathering derived ions such as $Na^+$, there was a slight increasing trend implying the
chemical inputs from the tributaries. These trends were in accordance with the lithology in the study
area. The carbonate is dominated in the upper stream basin, when river drainages this area, carbonate
weathering contributes to the elevation of $Ca^{2+}$ and $HCO_3^-$. As the river entered into the down stream
dominated with silicate, the relative low ion concentrations due to silicate weathering contributed
to dilute the $Ca^{2+}$ and introduce extra $Na^+$ in the river.

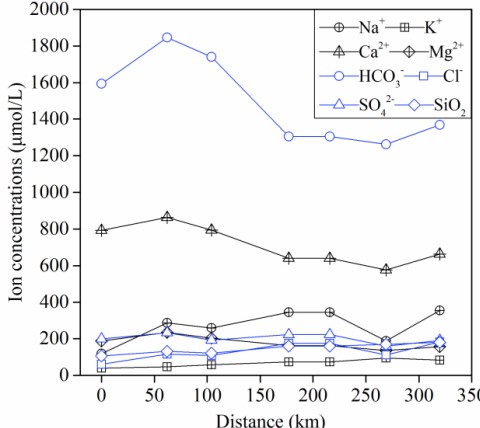


**Fig. 4 Spatial variations of major ion and SiO₂ concentrations in the Beijiang River Basin (From**
**upstream station CXs to the downstream station SJs)**



**5    Discussion**
**5.1  Chemical weathering rates and the controlling factors**
**5.1.1      Chemical weathering rates**
Atmospheric precipitation inputs, anthropogenic inputs and chemical weathering of rocks and
minerals as the major sources contributed to the hydrochemistry in the river basin. Previous studies
have shown that rock weathering contributions to major element composition of the river can be
interpreted in terms of mixing between three main end-members corresponding to the weathering
products of carbonates, silicates and evaporates (Cao et al., 2016b; Négrel et al., 1993; Ollivier et
al., 2010). The river water samples in the Beijiang River Basin were displayed on the plots of Na-
normalized molar ratios (Fig. 5). The best correlations were observed between $Ca^{2+}/Na^+$ and
$Mg^{2+}/Na^+$ ($R^2$=0.86, n=180) and $Ca^{2+}/Na^+$ and $HCO_3^-/Na^+$ ($R^2$=0.96, n=180). In these plots, the
contributions from carbonate weathering correspond to the trend toward high-$Ca^{2+}$ end-member
close to the top right corner, while silicate weathering correspond to the trend toward to high-$Na^+$
end-member close to the low-left corner. It was clear that the samples with high ratio of carbonate
outcrop had the highest molar ratios of $Ca^{2+}/Na^+$, $Mg^{2+}/Na^+$ and $HCO_3^-/Na^+$, which made the
samples located toward to the carbonate weathering end-member. However, the samples with low
$Ca^{2+}/Na^+$, $Mg^{2+}/Na^+$ and $HCO_3^-/Na^+$ ratios showed the influence of silicate weathering. In addition,
major ion compositions of the Beijiang River was mainly contributed by the weathering of
carbonates and silicates, and showed little contribution of evaporate weathering.


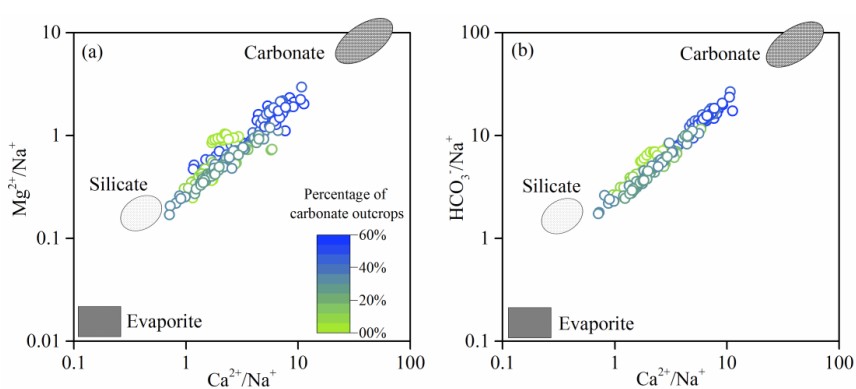

**Fig. 5 Mixing diagrams using Na-normalized molar ratios: (a) $Mg^{2+}/Na^+$ vs. $Ca^{2+}/Na^+$ (b) $HCO_3^-$/Na$^+$ vs. $Ca^{2+}/Na^+$ for the Beijiang River Basin. The color ramp showed the percentage of carbonate outcrops**

Based on the chemical balance method, the calculated contributions of different sources to the total cationic loads were presented in Fig. 6. The results showed that carbonate weathering was the most important mechanism controlling the local hydrochemistry, and contributed approximately 50.06% (10.96%~79.96%) of the total cationic loads. Silicate weathering and atmospheric precipitation inputs accounted for 25.71% (5.55%~70.38%) and 17.92% (0~46.95%), respectively. Evaporate weathering had the minimum contribution with an average of 6.31% (0~24.36%) to the total cationic loads.

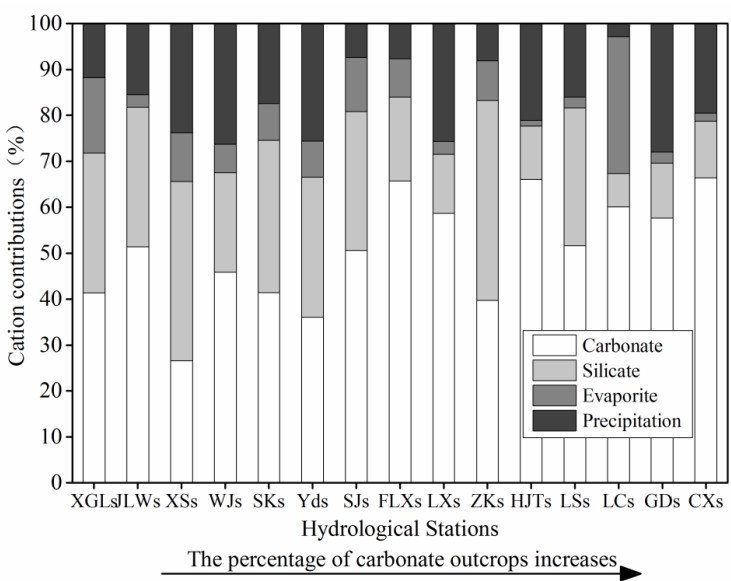


**Fig. 6 Calculate contributions (in %) from the different hydrological stations to the total cationic**

**load in the Beijiang River Basin. The cationic loads were the sum of $Na^+$, $K^+$, $Ca^{2+}$ and $Mg^{2+}$**

The chemical weathering rates were calculated and the results were listed in Table 2. The

average of carbonate and silicate weathering rate in the Beijiang River Basin were 61.15 and 25.31
$t\cdot km^{-2}\cdot a^{-1}$, respectively. In addition, chemical weathering rates showed significantly seasonal
variations with the highest carbonate and silicate weathering rates in May (16.75 and 5.50 $t\cdot km^{-2}\cdot month^{-1}$, respectively) and the lowest carbonate and silicate weathering rates in February (0.95 and
0.39 $t\cdot km^{-2}\cdot month^{-1}$, respectively). (Gaillardet et al., 1999a) reported the chemical weathering rate
of major river all over the world. The summarized dataset was showed in Fig. 7. It is found that the
hyperactive zone with high chemical weathering rate is generally located between the latitude 0-30°
and our study is belong to this area. Average CWR and SWR were about 61.15 and 25.31 $t\cdot km^{-2}\cdot month^{-1}$, respectively. The carbonate weathering contribute about 70% of the total chemical
weathering. The factors influence the balance between CWR and SWR would be further discussed





in the following parts.

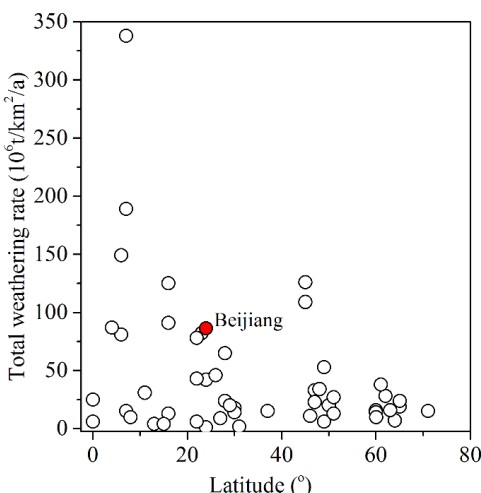


**Fig. 7 Relationship between latitude and total weathering rate (TWR)**


**Table 2 The annual discharge, catchment area, carbonate and silicate outcrops proportions, and**
**calculated weathering rates of carbonate and silicate of 15 subcatchments in the Beijiang River**

| ID | Annual discharge ($10^8$ m³/a) | Catchment area (km²) | Percentages of carbonate (%) | Percentages of silicate (%) | Carbonate weathering rate -CWR (t km⁻² year⁻¹) | Silicate weathering rate -SWR (t km⁻² year⁻¹) | Total weathering rate -TWR (t km⁻² year⁻¹) |
|---|---|---|---|---|---|---|---|
| JLWs | 2.23 | 281.13 | 2.95 | 97.05 | 18.63 | 14.94 | 33.56 |
| CXs | 4.06 | 392.35 | 57.44 | 42.56 | 74.21 | 11.42 | 85.64 |
| HJTs | 11.54 | 503.02 | 41.99 | 55.83 | 169.12 | 29.73 | 198.85 |
| ZKs | 16.38 | 1655.22 | 34.60 | 61.81 | 35.03 | 24.14 | 59.17 |
| XGLs | 13.56 | 1863.02 | 0.38 | 93.07 | 25.75 | 13.96 | 39.72 |
| WJs | 19.11 | 1960.99 | 12.51 | 73.87 | 55.00 | 17.43 | 72.43 |
| LXs | 56.37 | 2458.06 | 34.32 | 64.07 | 178.71 | 29.39 | 208.10 |
| LCs | 58.74 | 5278.14 | 49.67 | 50.21 | 79.70 | 20.59 | 100.29 |
| LSs | 74.83 | 6994.69 | 44.59 | 52.44 | 69.28 | 14.94 | 84.22 |
| XSs | 62.11 | 7497.01 | 7.09 | 87.81 | 18.85 | 20.35 | 39.20 |
| GDs | 137.81 | 9028.38 | 49.93 | 44.93 | 111.73 | 19.19 | 130.92 |
| SKs | 49.51 | 17417.24 | 25.43 | 69.35 | 12.71 | 6.11 | 18.82 |
| YDs | 191.07 | 18234.64 | 25.63 | 68.05 | 52.37 | 19.59 | 71.95 |





| | | | | | | | |
|---|---|---|---|---|---|---|---|
| FLXs | 396.25 | 34232.34 | 29.68 | 63.49 | 68.38 | 17.53 | 85.91 |
| SJs(Average) | 450.90 | 38538.06 | 28.12 | 64.65 | 61.15 | 25.31 | 86.46 |


### 5.1.2    Factors affecting chemical weathering

Many factors control the chemical weathering rates within river basins, including terrain,
geotectonic properties, lithology, land cover, climatic conditions (temperature, precipitation, etc.),
and hydrological characteristics (Ding et al., 2017; Gislason et al., 2009; Hagedorn and Cartwright,
2009). For this study, the lithology, hydrological characteristics and geomorphology was selected
as the major factors to be discussed.

### 5.1.2.1    Lithology

Among all the factors controlling the chemical weathering rates, lithology is one of the most
important factors because different type of rocks have different weathering abilities (Viers et al.,
2014). The TWR had a significant positive correlation ($p<0.01$) with the ratios of the proportion of
carbonate and a non significant positive correlation with that of silicate outcrops (Fig. 8a, b). Further
more, a significant correlation ($p<0.01$) was found between the CWR and proportion of carbonate
outcrops (Fig. 8c), but the correlations between the SWR and the proportion of silicate outcrops
were low and not statistically significant ($p>0.05$, Fig. 8d). The correlation analysis confirmed that
carbonate outcrops ratios was the sensitive factor controlling the chemical weathering rates and the
rapid kinetics of carbonate dissolution played an important role in weathering rates in the Beijiang
River Basin.



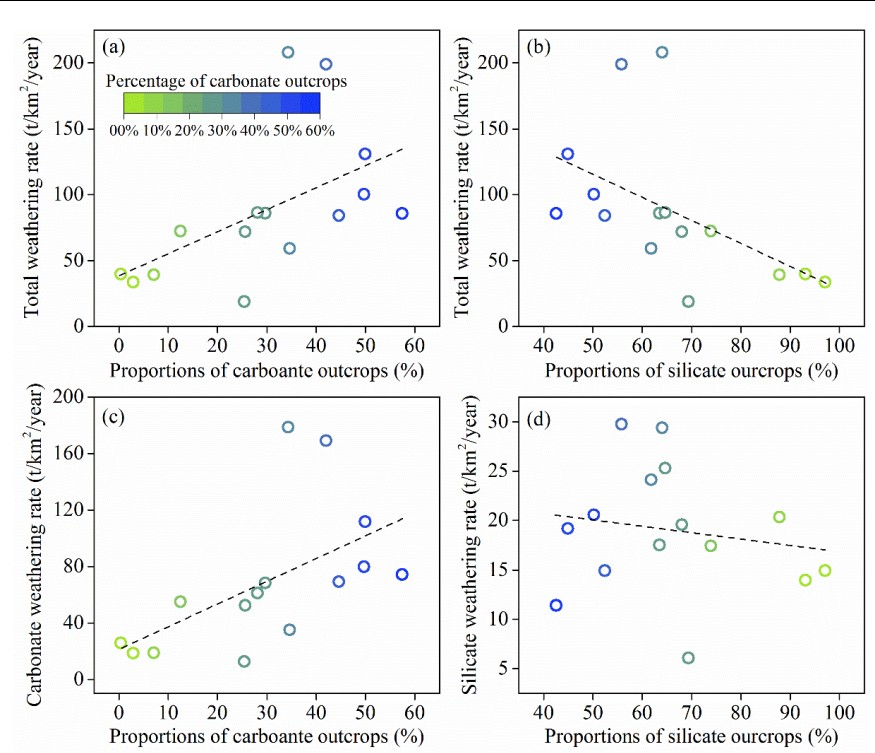


**Fig. 8 The relationships between weathering rates and the proportions of carbonate or silicate**

**outcrops**

**5.1.2.2  Runoff**
Chemical weathering is a combination of two processes, including dissolution of primary
minerals/glasses and precipitation of secondary minerals/biota growth (Eiriksdottir et al., 2011;
Hartmann et al., 2014a; Liu et al., 2013). The dissolution process is quite related to the precipitation
and runoff. In general, river water chemistry is usually diluted by river runoff (Q), and this dilution
effect is variable in different basins (Rao et al., 2019). The dilution effects of major element caused
by increasing water flow can be expressed by following log linear equation, the standard rating
relationship (Li et al., 2014; Walling, 1986; Zhang et al., 2007a):
$$C_i = aQ^b$$



where $C_i$ is the concentration of element i (mmol/L), Q is the water discharge (m$^3$/s), a is the
regression constant and b is the regression exponent. The linear fitting result was showed by Fig. 9
and the parameters b for major elements obtained from the dataset were 0.08 (Na$^+$), 0.05 (K$^+$), 0.08
(Ca$^{2+}$), 0.02 (Mg$^{2+}$), 0.06 (HCO$_3^-$), 0.12 (Cl$^-$), 0.11 (SO$_4^{2-}$) and -0.005 (SiO$_2$), respectively. In many
cases, b ranges from -1 to 0 due to the chemical variables that are influenced in various ways and
various extents. However, in our study area, the values of b were positive and not comparable to the
observations in the major Asian River such as the Yangtze (Chen et al., 2002a), the Yellow (Chen et
al., 2005), the Pearl Rivers (Zhang et al., 2007a) and the Mekong River (Li et al., 2014). This
suggests additional and significant solute sources in the river basin that may contribute and
compensate considerably the effect of dilution by precipitation. The difference of slope for
individual dissolved components at different stations reflects the different sources and the solubility
of source materials.

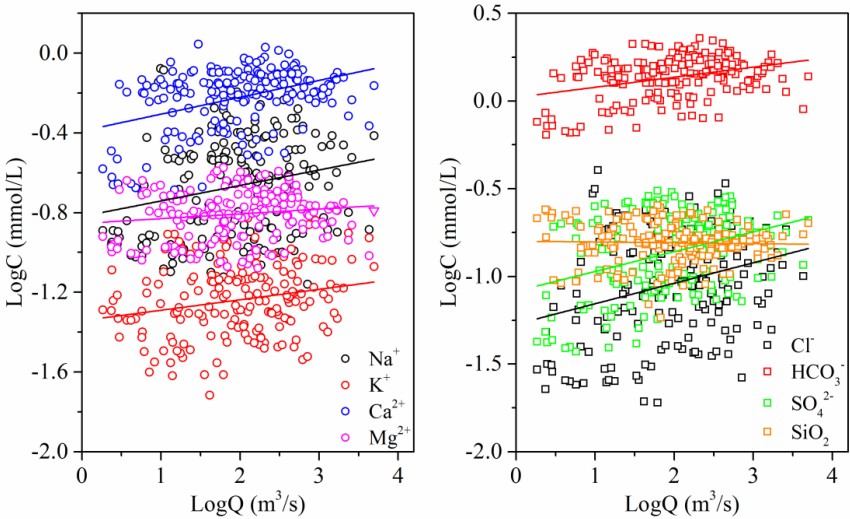


**Fig. 9 The relationship between major ion concentrations and runoff (Q) in logarithmic scales**

Due to the compensation effect of chemical weathering, significant positive linear relationship





was detected between Q and TWR, CWR and SWR. So that, the linear regression analysis between
Q and TWR, CWR and SWR were conducted to further reveal the effect of runoff on chemical
weathering rate. The slope of the liner regression equations for all 15 hydrological station
watersheds in the Beijiang River Basin were summarized in Table 3. The linear relations indicated
that the increase of runoff could accelerate the chemical weathering rates, but the variations of K
values revealed that the degrees of influences were different due to multiple factor influence, such
as the influence of geomorphology.
**Table 3 The slope of the liner regression equation between runoff (Q) and total weathering rate**
**(TWR), carbonate weathering rate (CWR) and silicate weathering rate (SWR)**

| Hydrological stations | Total weathering rate =$K_1$Q | | Carbonate weathering rate =$K_2$Q | | Silicate weathering rate =$K_3$Q | |
|---|---|---|---|---|---|---|
| | $K_1$ | $R^2$ | $K_2$ | $R^2$ | $K_3$ | $R^2$ |
| JLWs | 0.3912 | 0.9983 | 0.2091 | 0.9962 | 0.1821 | 0.9993 |
| CXs | 0.6492 | 0.9335 | 0.5631 | 0.9250 | 0.0860 | 0.9378 |
| HJTs | 0.5117 | 0.9689 | 0.4421 | 0.9613 | 0.0695 | 0.9939 |
| ZKs | 0.0953 | 0.9679 | 0.0525 | 0.7612 | 0.0429 | 0.8037 |
| XGLs | 0.0835 | 0.9781 | 0.0558 | 0.9741 | 0.0278 | 0.9817 |
| WJs | 0.1017 | 0.9985 | 0.0842 | 0.9965 | 0.0175 | 0.8835 |
| LXs | 0.0968 | 0.9816 | 0.0843 | 0.9778 | 0.0125 | 0.9914 |
| LCs | 0.0486 | 0.8983 | 0.0401 | 0.8672 | 0.0085 | 0.9739 |
| LSs | 0.0359 | 0.9654 | 0.0286 | 0.9570 | 0.0073 | 0.9423 |
| XSs | 0.0180 | 0.9806 | 0.0080 | 0.9681 | 0.0100 | 0.9571 |
| GDs | 0.0252 | 0.9969 | 0.0216 | 0.9974 | 0.0036 | 0.9900 |
| SKs | 0.0116 | 0.9802 | 0.0083 | 0.9822 | 0.0033 | 0.9547 |
| Yds | 0.0106 | 0.9963 | 0.0081 | 0.9936 | 0.0026 | 0.9240 |
| FLXs | 0.0050 | 0.9681 | 0.0039 | 0.9485 | 0.0010 | 0.9949 |
| SJs | 0.0053 | 0.9883 | 0.0037 | 0.9706 | 0.0016 | 0.9778 |






### 5.1.2.3 Geomorphology


The geomorphology factors including catchment area, average slope and HI, which could quite
influence the runoff generation process and physical and chemical weathering, were selected to give
a further explanation of the variation of K values. As showed in Fig. 10a, the K values were found
a non-linear relationship with the areas of subcatchment and could be fitted by exponential decay
model, which showed that the K values decreased dramatically with the initial increasing of area
and quickly become stable after reaching the threshold. The threshold value for $K_1$, $K_2$ and $K_3$ was
about 5278 km$^2$. It indicated that the compensation effect was more significant in small catchment.
The average topographic slope of each subcatchment ranged from 37° to 63°. With the
increasing of average slope, the residence time of both surface water and groundwater decrease.
Kinetics of carbonate and silicate reactions were determined by the reaction time which could be
related by the residence time of water. In our study area, the K values showed non-linear negative
correlation with average slope (Fig. 10e, f, g). When the average slope increase, the resulted small
residence time (time of water-rock reactions) make the compensation effect also weak in the study
area.
Hypsometric analysis showed that the HI ranged from 0.18 to 0.34. According to the empirical
classification by HI (HI > 0.6, inequilibrium or young stage, 0.35 < HI ≤ 0.6, equilibrium or mature
stage, HI ≤ 0.35, monadnock or old age), the geomorphological development in the Beijiang River
was recognized as the old age, which reflect the erodible degree and erosion trend of the
geomorphology was high. Furthermore, the non-linear positive correlations between HI and K
values (Fig. 10g, h, i) also addressed that geomorphology development have significant influence
on chemical weathering and relating $CO_2$ consumption processes.

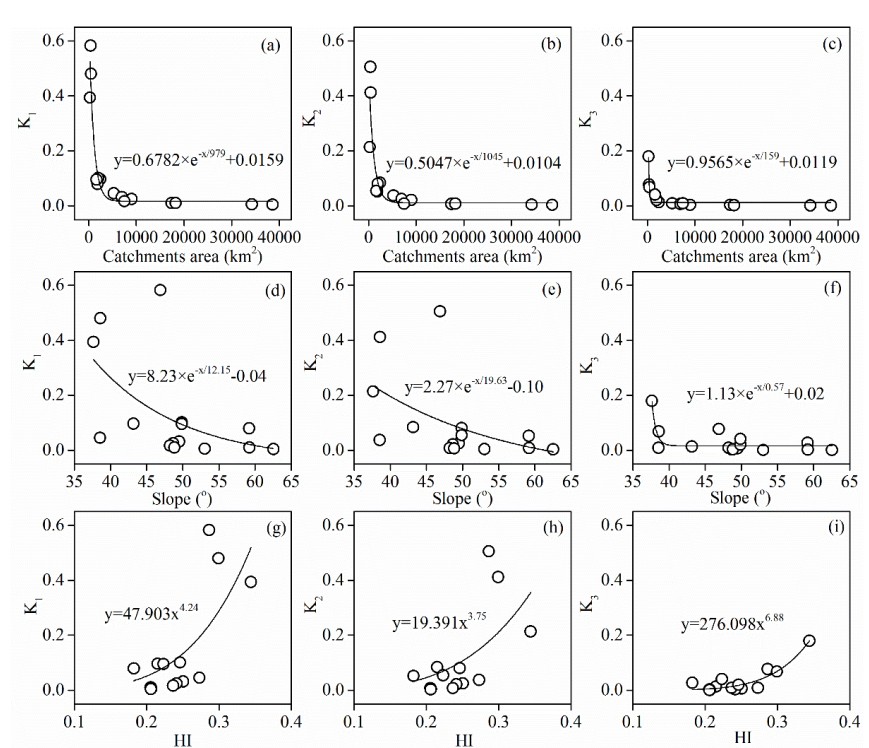

**Fig. 10 The relationships between K values and catchments area (a, b, c), average slope (d, e, f)**

**and HI (g, h, i) for the Beijiang River.**

**5.2  Temporary and net sink of atmospheric CO₂**

**5.2.1    Sulfate origin and DIC apportionment**

The successful application of DIC apportionment calculation mentioned in section 3.22 is

based on the sources of sulfate ($SO_4^{2-}$). Three sources of $SO_4^{2-}$ should be discriminated including

atmospheric acid deposition (Larssen and Carmichael, 2000), acid mining discharge (AMD) (Li et

al., 2018; Li et al., 2019) and chemical weathering of evaporite such as the dissolution of gypsum

(Appelo and Postma, 2005). Acid rain events occurred frequently in South and East China after

1980 (Larssen et al., 2006). The pH isolines based on data from 86 monitoring stations (Larssen et

al., 2006) showed that in the Beijiang River the rain pH was lower than 4.5 and our monitoring



dataset also proved this result. Sulfur wet deposition estimated based on the observed bulk wet sulfur
deposition data and the RAINS-Asia model (Larssen and Carmichael, 2000) ranged from 2000-
5000 eq ha$^{-1}$ a$^{-1}$, which showed that the acid sulfur deposition was one of the most important sources
of river sulfate. In addition, considering the abundant ore resources in the Beijiang River, the second
possible source of $SO_4^{2-}$ is sulfide oxidation due to mining. In our previous study, the $SO_4^{2-}$ with
AMD origin mainly came from the tributary Wenjiang River (Wen et al., 2018). These two sources
could offer sufficient chemical weathering agent $H_2SO_4$ and actively involved in the chemical
weathering due to the following reaction mechanism (take carbonate for example) (Taylor et al.,
1984; van Everdingen and Krouse, 1985).
$$FeS_2 + \frac{7}{2}O_2 + H_2O = Fe^{2+} + 2SO_4^{2-} + 2H^+$$
$$2CaCO_3 + H_2SO_4 = 2Ca^{2+} + 2HCO_3^- + SO_4^{2-}$$

The third source came from dissolution of gypsum could not offer active $H_2SO_4$ to induce

carbonate and silicate dissolution. Two evidence were summarized to indicate the absence of
gypsum in the study area, (1) Lithology in the river basin are composed of limestone, sandstone,
gneiss and glutenite. HI showed that geomorphology development have entered into the "old" age,
the evaporite such halite and gypsum have been consumed by the dissolution. (2) The stoichiometric
relationship between $Ca^{2+}$ and $SO_4^{2-}$ (Fig. 11) showed that all of the samples in the study area located
below the 1:1 gypsum dissolution line, and due to the dissolution of carbonate, they also below the
1:2 carbonate weathering induced by sulfuric acid (SCW) line. These two points combined gave the
evidence to prove the absence of contribution of gypsum dissolution to river $SO_4^{2-}$. So that, the DIC
apportionment could be calculated according to equation (18) to (21) and the result of three main
processes (CCW, CSW and SCW) contributing to the DIC origin in the Beijiang River water are
showed in Table 4. It was found that CCW was the dominant origin of DIC (35%~87%) and that
SCW (3%~15%) and CSW (7%~59%) were non-negligible weathering processes.

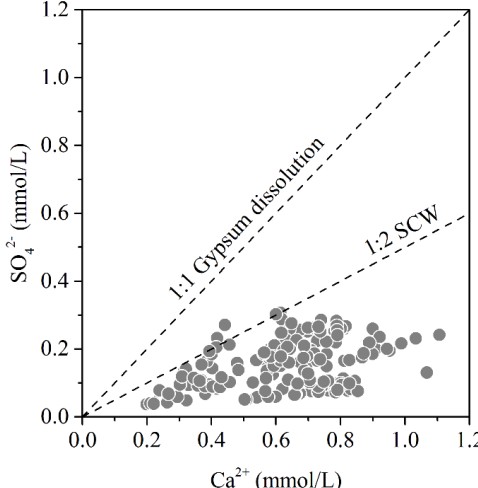

**Fig. 11 Stoichiometric relationship between $Ca^{2+}$ and $SO_4^{2-}$**
**5.2.2      Temporary and net $CO_2$ sink**

According to the classical view of the global carbon cycling (Berner and Kothavala, 2001),

the $CO_2$ sink induced by chemical weathering varies for different time scales. Aa short-term
timescale, carbonic acid based carbonate and silicate weathering (CCW and CSW) and transport of
the $HCO_3^-$ to oceans through rivers is an important "temporary" carbon sink (Khadka et al., 2014)
and can be calculated by the sum of $CCR_{CCW}$ and $CCR_{CSW}$. Thus, it was significant to estimate the
CCR of CCW and CSW (Liu and Dreybrodt, 2015; Liu et al., 2011). However, at the geological
timescale (>$10^6$ years), when over the timescale typical of residence time of $HCO_3^-$ in the ocean
($10^5$ years), the CCW is not a mechanism that can participate in the net sink of $CO_2$ in the atmosphere
because all of the atmospheric $CO_2$ fixed through CCW is returned to the atmosphere during
carbonate precipitation in the ocean. Meanwhile, in case of CSW, followed by carbonate deposition,



one of the two moles of $CO_2$ involved is transferred from the atmosphere to the lithosphere in the
form of carbonate rocks, while the other one returns to the atmosphere. The CSW is recognized as
the net sink of atmosphere $CO_2$. In addition, when sulfuric acid is involved as a proton donor in
carbonate weathering, half of the carbon dissolved to the atmospheric during carbonate precipitation.
Thus, SCW leads to a net release of $CO_2$ in ocean-atmosphere system. So that, the net $CO_2$ sink
(expressed by $CCR_{Net}$ in this study) is controlled by the DIC apportionment according to equation

(31).

The result of $CCR_{Total}$, $CCR_{CCW}$, $CCR_{CSW}$ and $CCR_{Net}$ were summarized in Table 4. The

$CCR_{Total}$ was 823.41 $10^3$ mol km$^{-2}$ a$^{-1}$. Comparing with other Chinese rivers, such as the Songhua
River ($189\times10^3$ mol km$^{-2}$ a$^{-1}$) (Cao et al., 2015) and other rivers calculated by (Gaillardet et al.,
1999a) including the Heilong River ($53\times10^3$ mol km$^{-2}$ a$^{-1}$), the Changjiang River ($609\times10^3$ mol km$^{-2}$ a$^{-1}$), the
$^2$ a$^{-1}$), the Huanghe River ($360\times10^3$ mol km$^{-2}$ a$^{-1}$), the Xijiang River ($960\times10^3$ mol km$^{-2}$ a$^{-1}$), the
Jinshajiang River ($420\times10^3$ mol km$^{-2}$ a$^{-1}$), the Langcangjiang River ($980\times10^3$ mol km$^{-2}$ a$^{-1}$), the
Nujiang River ($1240\times10^3$ mol km$^{-2}$ a$^{-1}$), the Yalongjiang River ($870\times10^3$ mol km$^{-2}$ a$^{-1}$), the Daduhe
River ($1280\times10^3$ mol km$^{-2}$ a$^{-1}$) and Minjiang River ($660\times10^3$ mol km$^{-2}$ a$^{-1}$), our study area showed
relative high CCR due to high chemical weathering rate. In addition, the $CCR_{CCW}$ and $CCR_{CSW}$ were
$536.59\times10^3$ (65%) and $286.82\times10^3$ (35%) mol km$^{-2}$ a$^{-1}$, respectively. Compared with the
"temporary" sink, the net sink of $CO_2$ for the Beijiang River was approximately $23.18\times10^3$ mol km$^{-2}$ a$^{-1}$ of $CO_2$ sinking in the perspective of global carbon cycling. It was about 3% of the "temporary"
$^2$ a$^{-1}$ of $CO_2$ sinking in the perspective of global carbon cycling. It was about 3% of the "temporary"
$CO_2$ sink. Human activities (sulfur acid deposition and AMD) dramatically decreased the $CO_2$ net
sink and even make chemical weathering a $CO_2$ source to the atmosphere.

**Table 4 Calculated $CO_2$ consumption rate and net sink of 15 nested subcatchments in the**





**Beijiang River Basin**

| Hydrological stations | DIC apportionment ($10^9$ mol/a) | | | "Temporary" Sink (CO$_2$ consumption rate) ($10^3$ mol km$^{-2}$ a$^{-1}$) | | | Net Sink ($10^3$ mol km$^{-2}$ a$^{-1}$) |
|---|---|---|---|---|---|---|---|
| | CCW | SCW | CSW | CCR$_{CCW}$ | CCR$_{CSW}$ | CCR$_{Total}$ | CCR$_{Net}$ |
| JLWs | 0.10 | 0.00 | 0.05 | 175.23 | 191.14 | 366.36 | 87.73 |
| CXs | 0.57 | 0.04 | 0.05 | 732.05 | 118.18 | 850.23 | 13.18 |
| HJTs | 1.57 | 0.06 | 0.34 | 1563.64 | 683.41 | 2247.05 | 286.14 |
| ZKs | 1.24 | 0.16 | 0.73 | 375.23 | 439.77 | 815.00 | 172.27 |
| XGLs | 0.85 | 0.14 | 0.37 | 227.05 | 195.91 | 422.95 | 61.59 |
| WJs | 1.76 | 0.17 | 0.87 | 449.32 | 443.18 | 892.50 | 177.50 |
| LXs | 7.30 | 0.40 | 2.61 | 1485.45 | 1060.45 | 2545.91 | 449.09 |
| LCs | 8.07 | 0.86 | 1.92 | 764.32 | 363.41 | 1127.95 | 99.77 |
| LSs | 10.13 | 0.42 | 2.48 | 724.55 | 354.32 | 1078.64 | 147.05 |
| XSs | 2.08 | 0.41 | 3.52 | 138.64 | 469.09 | 607.73 | 207.05 |
| GDs | 16.48 | 0.71 | 7.60 | 912.73 | 841.82 | 1754.55 | 381.36 |
| SKs | 4.00 | 0.72 | 1.74 | 114.77 | 100.23 | 215.00 | 29.55 |
| YDs | 14.11 | 1.75 | 13.10 | 386.82 | 718.64 | 1105.45 | 311.14 |
| FLXs | 40.38 | 7.74 | 4.46 | 589.77 | 130.45 | 720.23 | -47.73 |
| SJs | 41.36 | 9.27 | 11.05 | 536.59 | 286.82 | 823.41 | 23.18 |

## 6    Conclusions

This study revealed the temporary and net sinks of atmospheric CO$_2$ due to chemical

weathering in a subtropical hyperactive catchment with mixing carbonate and silicate lithology

under the stress of chemical weathering induced by anthropogenic sulfuric acid agent. During the

sampling period, the pH values ranged from 7.5 to 8.5 and TDS varied from 73.8 to 230.2 mg·L$^{-1}$.

Ca$^{2+}$ and HCO$_3^-$ were the dominated cation and anion. Water chemical patterns and PCA showed

that carbonate and silicate weathering were the most important processes controlling the local

hydrochemistry. In average, carbonate and silicate weathering contributed approximately 50.06%





and 25.71% of the total cationic loads, respectively.
The average of carbonate and silicate weathering rate in the Beijiang River Basin were 61.15
and 25.31 t·km$^{-2}$·a$^{-1}$, respectively. The high rate was comparable to other rivers located in the
hyperactive zone between the latitude 0-30º. The lithology, runoff and geomorphology had
significant influence on the chemical weathering rate. (1) Due to the difference between kinetics of
carbonate and silicate dissolution processes, the proportion of carbonate outcrops had significant
positive correlation with the chemical weathering rate and confirmed that carbonate outcrops ratios
was the sensitive factor controlling the chemical weathering rates and the rapid kinetics of carbonate
dissolution played an important role in weathering rates. (2) Runoff manly controlled the season
variations and the dilution effect was weak in the study area. Due to the compensation effect of
chemical weathering, significant positive linear relationship was detected between Q and TWR,
CWR and SWR. (3) The geomorphology factors such as slope and HI had non-linear correlation on
chemical weathering rate and showed significant scale effect, which revealed the complexity in
chemical weathering processes.
DIC apportionment showed that CCW was the dominant origin of DIC (35%-87%) and that
SCW (3%-15%) and CSW (7%-59%) were non-negligible weathering processes. The CCR$_{Total}$ was
823.41 $10^3$ mol km$^{-2}$ a$^{-1}$, relative high CCR due to high chemical weathering rate. In addition, the
CCR$_{CCW}$ and CCR$_{CSW}$ were 536.59×$10^3$ (65%) and 286.82×$10^3$ (35%) mol km$^{-2}$ a$^{-1}$, respectively.
Compared with the "temporary" sink, the net sink of CO$_2$ for the Beijiang River was approximately
23.18×$10^3$ mol km$^{-2}$ a$^{-1}$ of CO$_2$ sinking in the perspective of global carbon cycling. It was about
2.82% of the "temporary" CO$_2$ sink. Human activities induced sulfur acid deposition and AMD have
significantly alter the CO$_2$ sinks.



**7    Acknowledgments**
This research work was financially supported by the General Program of the National Natural
Science Foundation of China (No.41877470), the Natural Science Foundation of Guangdong
Province, China (No. 2017A030313231) and the Natural Science Foundation of Guangdong
Province, China (No. 2017A030313229).
**8    Code/Data availability:** Yes.
**9    Author contribution:** Cao Yingjie and Tang Changyuan designed the study, carried out the
field work, analyzed the results, and drafted the manuscript. Xuan Yingxue and Guan Shuai
participated in the field sampling and laboratory analysis. Peng Yisheng reviewed and edited the
original draft of the manuscript. All authors read and approved the final manuscript.
**10   Competing interests:** No.

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
