# Peer review of "Temporary and net sinks of atmospheric $CO_2$ due to chemical weathering in subtropical catchment with mixing carbonate and silicate lithology"

_Biogeosciences, 2019_

## Referee Comment (RC1) · Anonymous Referee #1 · 10 Dec 2019

Major Comments: The subject matter fits within the scope of the journal and the results are of interest to the readers. Chemical weathering is one of the major processes interacting with climate and tectonics to form clays, supply nutrients to soil microorganisms and plants, and sequester atmospheric CO2. The related researches are always the hot spots in global change. In the paper, the authors first discriminated carbonate weathering and silicate weathering by stoichiometric analysis based on mass balance. Then the DIC apportionments were applied to quantify the anthropogenic acid (major in from of sulfuric acid) contributions to chemical weathering. It is interesting the definition of temporary and net CO2 sinks. The primary findings are that of (1) Carbonate weathering dominated in the and contributed to about 70% the total weathering

rate. (2) The temporary CO2 sink was comparable to other subtropical basins. (3) The net sink was only 2.82% of the temporary sink and human activities dramatically decreased the CO2 net sink and even make chemical weathering a CO2 source to the atmosphere. The data analysis is for the most part sound, and the work does appear to be one of the first complete analysis of the chemical weathering and related CO2 consumption in this river.

Some questions: (1) During the calculation of chemical weathering rates, the authors ignore the anthropogenic origins of major ions except for $SO_4^{2-}$, show reasons (2) Part 5.1 is too long, may be it is a good idea to separate it into two parts. (3) Unify the reference format throughout the paper (4) Need to give more details about acid deposition and acid mining drainage (AMD) in the Beijiang River Basin Specific comments: Lines 36-37, "regulating the atmosphere-land-ocean fluxes and earth's climate" should be "regulating the atmosphere-land-ocean carbon fluxes and earth's climate" Lines 38-39, delete the "A profound case in point" Lines 54, delete "because" Lines 56, delete "(sulfide oxidation)" Lines 72-75, give some reference to this part Lines 93, change into "it covers an area of 52068 km2" Lines 129 delete "According to the principle of the mass balance" Lines 245 change into "chemical compositions" Lines 282-284, "Nov", "Jun" and "Feb" should give full names. Line 289, "It is" should be "It was". Lines 450-451, Equations are not labeled a Eq. number.Lines 466, "SCW" should give a explain in the Fig. 11 Line 485, "The result of CCRTotal, CCRCCW, CCRCSW and CCRNET were summarized in Table 4" should be "The results of CCRTotal, CCRCCW, CCRCSW and CCRNET are summarized in Table 4". Line 514, "significant influence" should be "significant influences". Line 518, "Runoff manly controlled" should be "Runoff mainly controlled". Lines 530-531, How human activities induced sulfur acid deposition altered the CO2 sinks, increased or decreased?

---

## Referee Comment (RC2) · Anonymous Referee #2 · 10 Jan 2020

The authors investigated major ion chemistry in Beijiang river water in China and calculated chemical weathering and CO2 consumption rate in the basin. They distinguished chemical weathering of silicate and carbonate and their agents; carbonic and sulfuric acid. I agree that some previous studies have ignored "anthropogenic" weathering and this difference is important to elucidate the global carbon cycle on different timescales, about which the authors used phrases of "temporary and net sink of atmospheric CO2". The subject of this paper is good for the journal.

I understand that the authors have collected abundant data in different sampling stations and seasons. However, I have serious concerns over the description of the data

and calculation methods. For example, the mass and chemical parameters of rainwater are not provided, and I couldn't assess the results. There are no information about analytical errors. The authors seem to confuse alkalinity, DIC, and [$HCO_3-$], which have totally different definitions (although I understand that these parameters are similar at pH 8 in the river waters, $HCO_3-$ is the main topic of this paper and the authors should calculate and explain accurately). Are the chemical parameters of the river (and relevant calculation results) weighted average over 12 months? What kind of methods do the authors use to calculate the area of silicate/carbonate outcrops or river water discharge?

The background of this study is unclear, and the authors should provide more basic information. What is "hyperactive region"? I recognize that Beijiang River is a major tributary of the Pearl River, but this river is relatively small compared to other world major river such as Amazon or Changjiang River. How does this river contribute the global carbon cycle? In addition, I have no idea why the authors compared total chemical weathering rate with latitude.

Furthermore, there are also some previous studies about the Pearl River and its tributaries, some of which have already taken into consideration anthropogenic weathering in some way. Do the author's $HCO_3-$-basis calculation methods and their results make a difference? I think the last section in discussion is too descriptive. I also have a concern that temporary and net sink of CO2 show large spatial variations, but in the discussion, the authors mentioned these values only in the SJs station (lowermost part).

Overall, the data and subject of this paper are good, but I'm regret to say that there are many problems for the description. At this stage I couldn't recommend publication of this paper.

Question: as shown in equation (21), silicate weathering by sulfuric acid does not affect the concentrations of $HCO_3-$ in the river. However, in equation (23) and (24),

[SO42−]ssw seemed to be described as $\alpha$CSW×$\alpha$SCW /$\alpha$CCW × [HCO3−]riv. Would you please explain this calculation?

---

## Author Comment (AC1) · 10 Feb 2020

Responses to Anonymous Referee #1:

Thank you for your time and sincere evaluation for our manuscript. Thank you very much for your constructive comments, and they are very useful for improving our manuscript. We have revised the manuscript according to the suggestions and comments, and the responses to questions one by one are as follows.

Responses to the questions:

Question 1: During the calculation of chemical weathering rates, the authors ignore the

anthropogenic origins of major ions except for SO42-, show reasons.

Answer 1:Answer 1: Thank you for your question. There are two reasons. (1) Two main characteristics of much polluted rivers are that TDS is greater than 500 mg/L and the Cl-/Na+ molar ratio is greater than that of sea salts (about 1.16) (Cao et al., 2016; Gaillardet et al., 1999). The TDS in the study area ranged from 73.79 to 230.16 mgÅůL-1 and the low TDS implied that the anthropogenic origins of major ions could be ignored in the study. The Beijiang River is characterized as a typical region suffered from serve acid deposition (Larssen et al., 2006) and active mining area (Li et al., 2019). The acid deposition and acid mining discharge contribute to the highest concentration of SO42-. (2) Natural origin of SO42- is the dissolution of evaporite, such as gypsum, while no evaporite was found in the study area. because if SO42- comes from the gypsum dissolution, the ratios of Ca2+ and SO42- should be close to 1:1. The stoichiometric analysis showed that the ratio between Ca2+ and SO42- deviated from 1:1 and also proved this point (Fig.11 in the manuscript and also showed Fig.1 as followed). The two reasons have been added in the lines 141-152.

Question 2: Part 5.1 is too long, may be it is a good idea to separate it into two parts.

Answer 2: Thank you for your suggestion. The part 5.1 has been separated into two parts: 5.1.1 Chemical weathering rates and 5.1.2 Factors affecting chemical weathering.

Question 3: Unify the reference format throughout the paper

Answer 3: Thank you for your suggestion. The format of reference has been modified throughout the paper.

Responses to the specific comments:

Comment 1: Lines 36-37, "regulating the atmosphere-land-ocean fluxes and earth's climate" should be "regulating the atmosphere-land-ocean carbon fluxes and earth's climate"

Answer 1: Thanks a lot. It has been modified in the lines 37-39.

Comment 2: Lines 38-39, delete the "A profound case in point"

Answer 2: Thanks a lot. It has been modified in the lines 41-44.

Comment 3: Lines 54, delete "because"

Answer 3: Thanks a lot. It has been modified in the lines 56-57.

Comment 4: Lines 56, delete "(sulfide oxidation)"

Answer 4: Thanks a lot. It has been modified in the lines 58-59.

Comment 5: Lines 93, change into "it covers an area of 52068 km2"

Answer 5: Thanks a lot. It has been modified in the lines 95-96.

Comment 6: Lines 129 delete "According to the principle of the mass balance"

Answer 6: Thanks a lot. It has been modified in the line 134.

Comment 7: Lines 245 change into "chemical compositions"

Answer 7: Thanks a lot. It has been modified in the lines 265.

Comment 8: Lines 282-284, "Nov", "Jun" and "Feb" should give full names.

Answer 8: Thanks a lot. It has been modified in the lines 305-308.

Comment 9: Line 289, "It is" should be "It was".

Answer 9: Thanks a lot. It has been modified in the lines 313-314.

Comment 10: Lines 450-451, Equations are not labeled a Eq. number.

Answer 10: Thanks a lot. It has been modified in the lines 472-473.

Comment 11: Lines 466, "SCW" should give a explain in the Fig. 11

Answer 11: Thanks a lot. It has been modified in the lines 489-490.

Comment 12: Line 485, "The result of CCRTotal, CCRCCW, CCRCSW and CCRNET were summarized in Table 4" should be "The results of CCRTotal, CCRCCW, CCRCSW and CCRNET are summarized in Table 4".

Answer 12: Thanks a lot. It has been modified in the lines 509.

Comment 13: Line 514, "significant influence" should be "significant influences".

Answer 13: Thanks a lot. It has been modified in the lines 544-545.

Comment 14: Line 518, "Runoff manly controlled" should be "Runoff mainly controlled".

Answer 14: Thanks a lot. It has been modified in the lines 549-550.

Comment 15: Lines 530-531, How human activities induced sulfur acid deposition altered the CO2 sinks, increased or decreased?

Answer 15: Thank you for your question. In addition to the chemical weathering induced by H2CO3, sulfuric acid (H2SO4) of anthropogenic origins produced by sulfide oxidation such as acid deposition caused by fossil fuel burning and acid mining discharge (AMD) also becomes an important chemical weathering agent in the catchment scale. Many studies have shown the importance of sulfide oxidation and subsequent dissolution of other minerals by the resulting sulfuric acid at catchment scale (Hercod et al., 1998; Spence and Telmer, 2005). Depending on the fate of sulfate in the oceans, sulfide oxidation coupled with carbonate dissolution could facilitate a release of CO2 to the atmosphere (Spence and Telmer, 2005), the carbonate weathering by H2SO4 plays a very important role in quantifying and validating the ultimate CO2 consumption rate.

Reference

Cao, Y., Tang, C., Cao, G., Wang, X., 2016. Hydrochemical zoning: natural and anthropogenic origins of the major elements in the surface water of Taizi River Basin, Northeast China. Environmental Earth Sciences, 75(9): 811. Gaillardet, J., Dupré,

[Figure]

B., Louvat, P., Allegre, C., 1999. Global silicate weathering and CO2 consumption rates deduced from the chemistry of large rivers. Chemical geology, 159(1-4): 3-30. Larssen, T., Lydersen, E., Tang, D., He, Y., Gao, J., Liu, H., Duan, L., Seip, H. M., Vogt, R. D., Mulder, J., Shao, M., Wang, Y., Shang, H., Zhang, X., Solberg, S., Aas, W., Okland, T., Eilertsen, O., Angell, V., Li, Q., Zhao, D., Xiang, R., Xiao, J., and Luo, J.: Acid Rain in China, Environmental Science & Technology, 40, 418-425, 10.1021/es0626133, 2006 Li, R., Tang, C., Li, X., Jiang, T., Shi, Y., and Cao, Y.: Reconstructing the historical pollution levels and ecological risks over the past sixty years in sediments of the Beijiang River, South China, Science of The Total Environment, 649, 448-460, 2019. Hercod, D. J., Brady, P. V., and Gregory, R. T.: Catchment-scale coupling between pyrite oxidation and calcite weathering, Chemical Geology, 151, 259-276, 1998. Spence, J., and Telmer, K.: The role of sulfur in chemical weathering and atmospheric CO2 fluxes: Evidence from major ions, $\delta$13CDIC, and $\delta$34SSO4 in rivers of the Canadian Cordillera, Geochimica et Cosmochimica Acta, 69, 5441-5458, 2005.

Please also note the supplement to this comment:
https://www.biogeosciences-discuss.net/bg-2019-310/bg-2019-310-AC1-supplement.pdf

―――――――――――――――――――――――

**Fig. 1.** Stoichiometric relationship between Ca2+ and SO42-

**Supplement:**

**Reply to interactive comments on "Temporary and net sinks of atmospheric CO₂ due to chemical weathering in subtropical catchment with mixing carbonate and silicate lithology" (bg-2019-310)**

**Responses to Anonymous Referee #1:**

Thank you for your time and sincere evaluation for our manuscript. Thank you very much for your constructive comments, and they are very useful for improving our manuscript. We have revised the manuscript according to the suggestions and comments, and the responses to questions one by one are as follows.

**Responses to the questions:**

**Question 1**: During the calculation of chemical weathering rates, the authors ignore the anthropogenic origins of major ions except for $SO_4^{2-}$, show reasons.

**Answer 1:** Thank you for your question.

There are two reasons.

(1) Two main characteristics of much polluted rivers are that TDS is greater than 500 mg/L and the $Cl^-/Na^+$ molar ratio is greater than that of sea salts (about 1.16) (Cao et al., 2016; Gaillardet et al., 1999). The TDS in the study area ranged from 73.79 to 230.16 $mg \cdot L^{-1}$ and the low TDS implied that the anthropogenic origins of major ions could be ignored in the study. The Beijiang River is characterized as a typical region suffered from serve acid deposition (Larssen et al., 2006) and active mining area (Li et al., 2019). The acid deposition and acid mining discharge contribute to the highest concentration of $SO_4^{2-}$.

(2) Natural origin of $SO_4^{2-}$ is the dissolution of evaporite, such as gypsum, while no evaporite was

found in the study area. because if $SO_4^{2-}$ comes from the gypsum dissolution, the ratios of $Ca^{2+}$ and $SO_4^{2-}$ should be close to 1:1. The stoichiometric analysis showed that the ratio between $Ca^{2+}$ and $SO_4^{2-}$ deviated from 1:1 and also proved this point (Fig.11 in the manuscript and also showed below).

The two reasons have been added in the lines 141-152.

[Figure]

Fig.11 Stoichiometric relationship between $Ca^{2+}$ and $SO_4^{2-}$

**Question 2**: Part 5.1 is too long, may be it is a good idea to separate it into two parts.

**Answer 2:** Thank you for your suggestion. The part 5.1 has been separated into two parts: 5.1.1 Chemical weathering rates and 5.1.2 Factors affecting chemical weathering.

**Question 3**: Unify the reference format throughout the paper

**Answer 3:** Thank you for your suggestion. The format of reference has been modified throughout the paper.

**Responses to the specific comments:**

**Comment 1:** Lines 36-37, "regulating the atmosphere-land-ocean fluxes and earth's climate" should be "regulating the atmosphere-land-ocean carbon fluxes and earth's climate"

**Answer 1:** Thanks a lot. It has been modified in the lines 37-39.

**Comment 2:** Lines 38-39, delete the "A profound case in point"

**Answer 2:** Thanks a lot. It has been modified in the lines 41-44.

**Comment 3:** Lines 54, delete "because"

**Answer 3:** Thanks a lot. It has been modified in the lines 56-57.

**Comment 4:** Lines 56, delete "(sulfide oxidation)"

**Answer 4:** Thanks a lot. It has been modified in the lines 58-59.

**Comment 5:** Lines 93, change into "it covers an area of 52068 km$^2$"

**Answer 5:** Thanks a lot. It has been modified in the lines 95-96.

**Comment 6:** Lines 129 delete "According to the principle of the mass balance"

**Answer 6:** Thanks a lot. It has been modified in the line 134.

**Comment 7:** Lines 245 change into "chemical compositions"

**Answer 7:** Thanks a lot. It has been modified in the lines 265.

**Comment 8:** Lines 282-284, "Nov", "Jun" and "Feb" should give full names.

**Answer 8:** Thanks a lot. It has been modified in the lines 305-308.

**Comment 9:** Line 289, "It is" should be "It was".

**Answer 9:** Thanks a lot. It has been modified in the lines 313-314.

**Comment 10:** Lines 450-451, Equations are not labeled a Eq. number.

**Answer 10:** Thanks a lot. It has been modified in the lines 472-473.

**Comment 11:** Lines 466, "SCW" should give a explain in the Fig. 11

**Answer 11:** Thanks a lot. It has been modified in the lines 489-490.

**Comment 12:** Line 485, "The result of $CCR_{Total}$, $CCR_{CCW}$, $CCR_{CSW}$ and $CCR_{NET}$ were summarized in Table 4" should be "The results of $CCR_{Total}$, $CCR_{CCW}$, $CCR_{CSW}$ and $CCR_{NET}$ are summarized in Table 4".

**Answer 12:** Thanks a lot. It has been modified in the lines 509.

**Comment 13:** Line 514, "significant influence" should be "significant influences".

**Answer 13:** Thanks a lot. It has been modified in the lines 544-545.

**Comment 14:** Line 518, "Runoff manly controlled" should be "Runoff mainly controlled".

**Answer 14:** Thanks a lot. It has been modified in the lines 549-550.

**Comment 15:** Lines 530-531, How human activities induced sulfur acid deposition altered the CO2 sinks, increased or decreased?

**Answer 15:** Thank you for your question. In addition to the chemical weathering induced by $H_2CO_3$, sulfuric acid ($H_2SO_4$) of anthropogenic origins produced by sulfide oxidation such as acid deposition caused by fossil fuel burning and acid mining discharge (AMD) also becomes an important chemical weathering agent in the catchment scale. Many studies have shown the importance of sulfide oxidation and subsequent dissolution of other minerals by the resulting sulfuric acid at catchment scale (Hercod et al., 1998; Spence and Telmer, 2005). Depending on the fate of sulfate in the oceans, sulfide oxidation coupled with carbonate dissolution could facilitate a release of $CO_2$ to the atmosphere (Spence and Telmer, 2005), the carbonate weathering by $H_2SO_4$ plays a very important role in quantifying and validating the ultimate $CO_2$ consumption rate.

**Reference**

Cao, Y., Tang, C., Cao, G., Wang, X., 2016. Hydrochemical zoning: natural and anthropogenic origins of the major elements in the surface water of Taizi River Basin, Northeast China. Environmental Earth Sciences, 75(9): 811.

Gaillardet, J., Dupré, B., Louvat, P., Allegre, C., 1999. Global silicate weathering and CO2 consumption rates deduced from the chemistry of large rivers. Chemical geology, 159(1-4): 3-30.

Larssen, T., Lydersen, E., Tang, D., He, Y., Gao, J., Liu, H., Duan, L., Seip, H. M., Vogt, R. D., Mulder, J., Shao, M., Wang, Y., Shang, H., Zhang, X., Solberg, S., Aas, W., Okland, T., Eilertsen, O., Angell, V., Li, Q., Zhao, D., Xiang, R., Xiao, J., and Luo, J.: Acid Rain in China, Environmental Science & Technology, 40, 418-425, 10.1021/es0626133, 2006

Li, R., Tang, C., Li, X., Jiang, T., Shi, Y., and Cao, Y.: Reconstructing the historical pollution levels and ecological risks over the past sixty years in sediments of the Beijiang River, South

China, Science of The Total Environment, 649, 448-460, 2019.

Hercod, D. J., Brady, P. V., and Gregory, R. T.: Catchment-scale coupling between pyrite oxidation and calcite weathering, Chemical Geology, 151, 259-276, 1998.

Spence, J., and Telmer, K.: The role of sulfur in chemical weathering and atmospheric $CO_2$ fluxes: Evidence from major ions, $\delta^{13}C_{DIC}$, and $\delta^{34}S_{SO4}$ in rivers of the Canadian Cordillera, Geochimica et Cosmochimica Acta, 69, 5441-5458, 2005.

---

## Author Comment (AC2) · 10 Feb 2020

Responses to Anonymous Referee #2:

Thank you for your time and sincere evaluation for our manuscript. Thank you very much for your constructive comments, and they are very useful for improving our manuscript. We have revised the manuscript according to the suggestions and comments, and the responses to questions one by one are as follows.

Question 1: I understand that the authors have collected abundant data in different sampling stations and seasons. However, I have serious concerns over the description

of the data and calculation methods. For example, the mass and chemical parameters of rainwater are not provided, and I couldn't assess the results.

Answer 1: Thank you very much for your suggestion about data of rainwater. We attach the major ions concentrations of rainwater in Table S1 in the supplementary material in the lines 139-140 and lines 799-800. We also present the data of rainwater here.

Question 2: There are no information about analytical errors. Answer 2: Thank you very much for your suggestion. Reference, blank and replicate samples were employed to check the accuracy of all the analysis and the relative standard deviations of all the analysis were within ±5%. The ionic charge balance defined by the equation of (meq(sum of cations)-meq(sum of anions))/(meq(sum of cations and anions)) of the water samples was less than 5%. The modified part was in the lines 125-131.

Question 3: The authors seem to confuse alkalinity, DIC, and [HCO3-], which have totally different definitions (although I understand that these parameters are similar at pH 8 in the river waters, HCO3- is the main topic of this paper and the authors should calculate and explain accurately).

Answer 3: Thank you very much for your question. The definitions of alkalinity, DIC, and [HCO3-] are different. The alkalinity describes the acid neutralizing capacity. It is determined by titrating with acid down to a pH of about 4.5. Equal to the concentrations of [HCO3-]+2[CO32-] (mmol/L) in most samples. DIC is the abbreviation of the dissolved inorganic carbon and is defined as the sum of [CO2] + [HCO3-]+[CO32-] in water samples. In this study the alkalinity is determined by titration in situ. The DIC which is defined as the sum of [CO2]+ [HCO3-]+[CO32-] can be calculated by using the [HCO3-], water temperature (T) and pH measured in the field according to the equations in the Supplement file. In addition, for all the samples, the pH values ranged from 7.5 to 8.5 with an average of 8.05. Under this pH conditions, the major species of DIC is HCO3-. Based on our calculation, H2CO3* and CO32- only account for less than 5% in all sampling sites, so we use the concentrations of HCO3- (mmol/L) to represent

the DIC in this study.

Question 4: Are the chemical parameters of the river (and relevant calculation results) weighted average over 12 months?

Answer 4: Thank you very much for your question. In this study, the chemical parameters of river water in Table 1 in the paper were the flow-weighted average over 12 months. For every sampling station, the flow-weighted average of ion concentration can be expressed as equation in the Supplement file. Also, we add this information in the lines 266-271 in the manuscript. For the relevant calculation results, we did the calculations using month data and sum the month results to obtain the year result by equation (15), (16) and (17) in the lines 181-184 in the manuscript.

Question 5: What kind of methods do the authors use to calculate the area of silicate/carbonate outcrops or river water discharge?

Answer 5: Thank you very much for your question. The area of silicate/carbonate outcrops was calculated by hydrological module of ArcGIS based on geology map from provided by China Geological Survey. The data of river water discharge was provided by the local hydrology bureau. The information has been added in the lines 255-258.

Question 6: The background of this study is unclear, and the authors should provide more basic information. What is "hyperactive region"?

Answer 6: Thank you for your question. (1) Explanation of background: As described in the Introduction, from the view of the global carbon cycle, the $CO_2$ consumption due to carbonate weathering is recognized the "temporary" sink, while the consumption of $CO_2$ during the chemical weathering of silicate rocks has been regard as the net sink of $CO_2$ and regulates the global carbon cycle. Thus in carbonate-silicate mixing catchment, it is essential to distinguish proportions of the two most important lithological groups, i.e., carbonates and silicates, and evaluate the net $CO_2$ sink due to chemical weathering of silicate. In addition to the chemical weathering induced by $H_2CO_3$,

sulfuric acid (H2SO4) of anthropogenic origins produced by sulfide oxidation such as acid deposition caused by fossil fuel burning and acid mining discharge (AMD) also becomes an important chemical weathering agent in the catchment scale. Depending on the fate of sulfate in the oceans, sulfide oxidation coupled with carbonate dissolution could facilitate a release of CO2 to the atmosphere, the carbonate weathering by H2SO4 (sulfide oxidation) plays a very important role in quantifying and validating the ultimate CO2 consumption rate. Thus, under the influence of human activities, the combination of silicate weathering by H2CO3 and carbonate weathering by H2SO4 controlled the net sink of atmospheric CO2. The Pearl River includes three principal rivers: the Xijiang, Beijiang, and Dongjiang Rivers. The three river basins have distinct geological conditions. The Xijiang River is characterized as the carbonate-dominated area and the Dongjiang River has silicate as the main rock type. While the Beijiang River, which is the second largest tributary of the Pearl River, is characterized as a typical carbonate-silicate mixing basin. In addition, as the serve acid deposition and active mining area, chemical weathering induced by sulfuric acid make the temporary and net sink of atmospheric CO2 to be reevaluated. These two points make the study area is representative.

(2) About the "hyperactive region" According to the work of (Meybeck et al., 2006), the global coastal catchments were classified into eight classes based on the yields of riverine material by the COSCAT data set. In order to facilitate the visualization, mapping and comparison of river fluxes for any given material, the authors normalize all yields (Yi) to their global average (Y*). If the values of normalized yields Yi/Y* is between 5 and 10, the catchment is called the "hyperactive region". Based on the calculation (Meybeck et al., 2006), the Pearl River is the "hyperactive region".

Question 7: I recognize that Beijiang River is a major tributary of the Pearl River, but this river is relatively small compared to other world major river such as Amazon or Changjiang River. How does this river contribute the global carbon cycle?

Answer 7: Thank you very much for your question. Although the Beijiang River is

not as large as Amazon or Changjiang River, the study of chemical weathering and CO2 sink in the Beijiang River can represent the carbon source and sink of such a river basin to some extents. In addition, the information of chemical weathering and CO2 sink in the Beijiang River can also provide scientific evidence for global carbon cycle. The reasons why we chose the Beijiang River for our study area are that (1) The Beijiang River is characterized as a typical carbonate-silicate mixing basin, however, little study investigated chemical weathering and CO2 sink in such a mixing basin which has a different mechanism of chemical weathering compared to river basins with a simple lithology (carbonate or silicate dominant). (2) The Beijiang River is located in the subtropical area in South China, the warm and wet climatic conditions make the Beijiang River a hyperactive region in China. Water discharge and chemical weathering is highly seasonal due to the warm and humid summer monsoon and the cool and dry winter monsoon. (3) The Beijiang River is the second largest tributary of the Pearl River, and it covers a basin of 52 068 km2. The study of chemical weathering and CO2 sink of the Beijiang River Basin is a supplement to the study of carbon cycle of the Pearl River which is the second largest river in China in terms of discharge volume.

Question 8: In addition, I have no idea why the authors compared total chemical weathering rate with latitude.

Answer 8: Thank you very much for your question. Based on the work of (Meybeck et al., 2006) and other researchers, the chemical weathering rate shows significant spatial trend. Generally it is found that the riverine output of materials is large in the low latitude area due to large runoffs . So in this study, we compared total chemical weathering rate with latitude to give further evidence to support the conclusion.

Question 9: Furthermore, there are also some previous studies about the Pearl River and its tributaries, some of which have already taken into consideration anthropogenic weathering in some way. Do the author's HCO3−-basis calculation methods and their results make a difference?

Answer 9: Thank you very much for your question. Based on our calculation method, the results in this study have compared with other Chinese rivers, as well as the Xijiang River which is the largest tributary of the Pearl River (see Lines 486-492 in Section 5.2.2). The total of $CO_2$ consumption rates CCR was $823.41 \times 10^3$ mol km-2 a-1 in the Beijiang River and was $960 \times 10^3$ mol km-2 a-1 in the Xijiang River. The total of $CO_2$ consumption rates in our study area showed little lower than that in the Xijiang River of the previous study.

In addition, some previous studies calculated the DIC apportionment based on the carbon isotope of DIC, however, our study calculated the DIC apportionment based on mass balance and $HCO_3^-$ concentration, the difference of these two methods will discuss in our other paper. Actually, this manuscript is focused on (1) the chemical weathering rate and the controlling factors on chemical weathering processes, and (2) the temporary sink of $CO_2$ and the influence of sulfide oxidation on net sink of $CO_2$ by DIC apportionment procedure. Thank you very much for your attention to our studies, we hope our study can provide further information for global carbon cycle studies.

Question 10: I think the last section in discussion is too descriptive. I also have a concern that temporary and net sink of $CO_2$ show large spatial variations, but in the discussion, the authors mentioned these values only in the SJs station (lowermost part).

Answer 10: Thank you very much for your suggestion. Actually, SJs station is the lowest station of the Beijiang River, which can represent the temporary and net sink of $CO_2$ of the whole river basin. In addition, the $CO_2$ net sink of each sub basin were also different and show large spatial variations due to heterogeneity of geology and human activities. The geology showed weak correlation with the $CO_2$ net sink (Fig. 1a), while the $SO_4^{2-}$ have negative correlation with the $CO_2$ net sink (Fig. 1b). It proved that human activities (sulfur acid deposition and AMD) dramatically decreased the $CO_2$ net sink and even make chemical weathering a $CO_2$ source to the atmosphere. We have added this part in the lines 521-524.

Question 11: As shown in equation (21), silicate weathering by sulfuric acid does not affect the concentrations of HCO3- in the river. However, in equation (23) and (24), [SO42−]ssw seemed to be described as $\alpha$CSW×$\alpha$SCW /$\alpha$CCW × [HCO3-]riv. Would you please explain this calculation?

Answer 11: Thank you very much for your question. Firstly, we are very sorry for that there are two equations numbered (23). We have changed numbers in the revision manuscript. In other to explain clearly for this question, we present some of equations in the supplement file.

(1) If we do not use the equation (22) and (23) as followed, just two equations (24) and (26) can get based on mass balance, however, we have three unknowns ($\alpha$CCWïijŇ$\alpha$SCW and $\alpha$CSW). Thus, we have a hypothesis, according to the studies of (Galy and France-Lanord, 1999) and (Spence and Telmer, 2005), carbonate and silicate weathering by carbonic acid in the same ratio as carbonate and silicate weathering by sulfuric acid, the mass balance equations in the supplement file.

(2) According to the above equations (22) and (23), we can get a further equation (25) in the supplement file.

(3) Combing the equations (24), (25) and (26), the proportions of HCO3- derived from three end-members (CCW, SCW and CSW) can be calculated, and the DIC (equivalent to HCO3-) fluxes by different chemical weathering processes are calculated by equations in the supplement file.

Reference:

Appelo, C.A.J., Postma, D., 2004. Geochemistry, groundwater and pollution. CRC press. Meybeck, M., Dürr, H.H., Vörösmarty, C.J., 2006. Global coastal segmentation and its river catchment contributors: A new look at land‐ocean linkage. Global Biogeochemical Cycles, 20(1).

Please also note the supplement to this comment:
https://www.biogeosciences-discuss.net/bg-2019-310/bg-2019-310-AC2-supplement.pdf
* * *
[Figure]

**Fig. 1.** Correlations between CO2 net sinks and proportions of proportions of carbonate (a) and correlations between CO2 net sinks and SO42- (b)

**Supplement:**

**Reply to interactive comments on "Temporary and net sinks of atmospheric CO₂ due to chemical weathering in subtropical catchment with mixing carbonate and silicate lithology" (bg-2019-310)**

**Responses to Anonymous Referee #2:**

Thank you for your time and sincere evaluation for our manuscript. Thank you very much for your constructive comments, and they are very useful for improving our manuscript. We have revised the manuscript according to the suggestions and comments, and the responses to questions one by one are as follows.

**Question 1**: I understand that the authors have collected abundant data in different sampling stations and seasons. However, I have serious concerns over the description of the data and calculation methods. For example, the mass and chemical parameters of rainwater are not provided, and I couldn't assess the results.

**Answer 1:** Thank you very much for your suggestion about data of rainwater. We attach the major ions concentrations of rainwater in Table S1 in the supplementary material in the lines 139-140 and lines 799-800. We also present the data of rainwater here.

Table S1 The major ions concentrations of rain water samples at 5 hydrological stations in the Beijiang River (mean±SD).

| Hydrological stations | $Na^+$ (µmol/L) | $K^+$ (µmol/L) | $Ca^{2+}$ (µmol/L) | $Mg^{2+}$ (µmol/L) | $Cl^-$ (µmol/L) | $SO_4^{2-}$ (µmol/L) | $NO_3^-$ (µmol/L) |
|---|---|---|---|---|---|---|---|
| XGLs | 12.8±9.7 | 21.0±16.8 | 22.2±20.5 | 10.9±10.3 | 25.9±22.6 | 320.2±370.7 | 83.3±85.2 |
| XSs | 20.4±11.8 | 7.8±4.5 | 86.9±30.4 | 10.1±5.2 | 10.0±0.0 | 606.5±511.5 | 36.3±23.4 |
| Yds | 16.3±9.5 | 10.1±10.8 | 161.1±56.5 | 9.0±7.8 | 23.9±12.4 | 136.9±169.5 | 143.1±135.5 |
| FLXs | 18.8±12.3 | 3.2±2.5 | 31.1±17.7 | 4.2±2.7 | 23.1±16.6 | 45.4±27.5 | 77.1±70.4 |
| SJs | 12.6±9.2 | 12.5±16.3 | 22.9±13.8 | 15.4±18.1 | 25.4±16.0 | 79.0±79.8 | 156.7±206.4 |

**Question 2**: There are no information about analytical errors.

**Answer 2:** Thank you very much for your suggestion. Reference, blank and replicate samples were employed to check the accuracy of all the analysis and the relative standard deviations of all the analysis were within ±5%. The ionic charge balance defined by the equation of $\frac{meq(sum\ of\ cations)-meq(sum\ of\ anions)}{meq(sum\ of\ cations\ and\ anions)}$ of the water samples was less than 5%. The modified part was in the lines 125-131.

**Question 3**: The authors seem to confuse alkalinity, DIC, and [HCO$_3^-$], which have totally different definitions (although I understand that these parameters are similar at pH 8 in the river waters, HCO$_3^-$ is the main topic of this paper and the authors should calculate and explain accurately).

**Answer 3:** Thank you very much for your question. The definitions of alkalinity, DIC, and [HCO$_3^-$] are different. The alkalinity describes the acid neutralizing capacity. It is determined by titrating with acid down to a pH of about 4.5. Equal to the concentrations of [HCO$_3^-$]+2[CO$_3^{2-}$] (mmol/L) in most samples. DIC is the abbreviation of the dissolved inorganic carbon and is defined as the sum of [CO$_2$] + [HCO$_3^-$]+[CO$_3^{2-}$] in water samples. In this study the alkalinity is determined by titration in situ.

The DIC which is defined as the sum of [CO$_2$]+ [HCO$_3^-$]+[CO$_3^{2-}$] can be calculated by using the [HCO$_3^-$], water temperature (T) and pH measured in the field according to the equation as follows:

$$H_2CO_3^* \leftrightarrow H^+ + HCO_3^-$$

$$HCO_3^- \leftrightarrow 2H^+ + CO_3^{2-}$$

$$K_1 = \frac{[H^+] \times [HCO_3^-]}{[H_2CO_3^*]} = 10^{(-1.1 \times 10^{-4} \times T^2 - 0.012T - 6.58)}$$

$$K_2 = \frac{[H^+] \times [CO_3^{2-}]}{[HCO_3^-]} = 10^{(-9 \times 10^{-5} \times T^2 + 0.0137T - 10.62)}$$

In addition, for all the samples, the pH values ranged from 7.5 to 8.5 with an average of 8.05. Under this pH conditions, the major species of DIC is $HCO_3^-$ (Fig.C1). Based on our calculation, $H_2CO_3^*$ and $CO_3^{2-}$ only account for less than 5% in all sampling sites, so we use the concentrations of $HCO_3^-$ (mmol/L) to represent the DIC in this study.

[Figure]

Fig.C1 Percentage of $HCO^{3-}$ of total dissolved carbonate as function of pH (Appelo and Postma, 2004)

Question 4: Are the chemical parameters of the river (and relevant calculation results) weighted average over 12 months?

Answer 4: Thank you very much for your question. In this study, the chemical parameters of river water in Table 1 in the paper were the flow-weighted average over 12 months. For every sampling station, the flow-weighted average of ion concentration can be expressed as followed equation:

$$[X]_{avarage} = \frac{\sum_{i=1}^{n=12} [X]_i \times Q_i}{\sum_{i=1}^{n=12} Q_i}$$

Where [X] is denotes the elements of the elements of $Ca^{2+}$, $Mg^{2+}$, $Na^+$, $K^+$, $Cl^-$, $SO_4^{2-}$, $HCO_3^-$ in mmol·L$^{-1}$. Q denotes average monthly discharge in m$^3$·s$^{-1}$. The subscripts i denotes 12 moths from January to December. Also, we add this information in the lines 266-271 in the manuscript. For

the relevant calculation results, we did the calculations using month data and sum the month results to obtain the year result by equation (15), (16) and (17) in the lines 181-184 in the manuscript.

**Question 5**: What kind of methods do the authors use to calculate the area of silicate/carbonate outcrops or river water discharge?

**Answer 5:** Thank you very much for your question. The area of silicate/carbonate outcrops was calculated by hydrological module of ArcGIS based on geology map from provided by China Geological Survey. The data of river water discharge was provided by the local hydrology bureau. The information has been added in the lines 255-258.

**Question 6**: The background of this study is unclear, and the authors should provide more basic information. What is "hyperactive region"?

**Answer 6:** Thank you for your question.

(1) Explanation of background: As described in the Introduction, from the view of the global carbon cycle, the $CO_2$ consumption due to carbonate weathering is recognized the "temporary" sink, while the consumption of $CO_2$ during the chemical weathering of silicate rocks has been regard as the net sink of $CO_2$ and regulates the global carbon cycle. Thus in carbonate-silicate mixing catchment, it is essential to distinguish proportions of the two most important lithological groups, i.e., carbonates and silicates, and evaluate the net $CO_2$ sink due to chemical weathering of silicate. In addition to the chemical weathering induced by $H_2CO_3$, sulfuric acid ($H_2SO_4$) of anthropogenic origins produced by sulfide oxidation such as acid deposition caused by fossil fuel

burning and acid mining discharge (AMD) also becomes an important chemical weathering agent in the catchment scale. Depending on the fate of sulfate in the oceans, sulfide oxidation coupled with carbonate dissolution could facilitate a release of $CO_2$ to the atmosphere, the carbonate weathering by $H_2SO_4$ (sulfide oxidation) plays a very important role in quantifying and validating the ultimate $CO_2$ consumption rate. Thus, under the influence of human activities, the combination of silicate weathering by $H_2CO_3$ and carbonate weathering by $H_2SO_4$ controlled the net sink of atmospheric $CO_2$.

The Pearl River includes three principal rivers: the Xijiang, Beijiang, and Dongjiang Rivers. The three river basins have distinct geological conditions. The Xijiang River is characterized as the carbonate-dominated area and the Dongjiang River has silicate as the main rock type. While the Beijiang River, which is the second largest tributary of the Pearl River, is characterized as a typical carbonate-silicate mixing basin. In addition, as the serve acid deposition and active mining area, chemical weathering induced by sulfuric acid make the temporary and net sink of atmospheric $CO_2$ to be reevaluated. These two points make the study area is representative.

(2) About the "hyperactive region"

According to the work of (Meybeck et al., 2006), the global coastal catchments were classified into eight classes based on the yields of riverine material by the COSCAT data set. In order to facilitate the visualization, mapping and comparison of river fluxes for any given material, the authors normalize all yields (Yi) to their global average (Y*). If the values of normalized yields Yi/Y* is between 5 and 10, the catchment is called the "hyperactive region". Based on the calculation (Meybeck et al., 2006), the Pearl River is the "hyperactive region".

**Question 7**: I recognize that Beijiang River is a major tributary of the Pearl River, but this river is relatively small compared to other world major river such as Amazon or Changjiang River. How does this river contribute the global carbon cycle?

**Answer 7:** Thank you very much for your question. Although the Beijiang River is not as large as Amazon or Changjiang River, the study of chemical weathering and CO2 sink in the Beijiang River can represent the carbon source and sink of such a river basin to some extents. In addition, the information of chemical weathering and $CO_2$ sink in the Beijiang River can also provide scientific evidence for global carbon cycle. The reasons why we chose the Beijiang River for our study area are that (1) The Beijiang River is characterized as a typical carbonate-silicate mixing basin, however, little study investigated chemical weathering and $CO_2$ sink in such a mixing basin which has a different mechanism of chemical weathering compared to river basins with a simple lithology (carbonate or silicate dominant). (2) The Beijiang River is located in the subtropical area in South China, the warm and wet climatic conditions make the Beijiang River a hyperactive region in China. Water discharge and chemical weathering is highly seasonal due to the warm and humid summer monsoon and the cool and dry winter monsoon. (3) The Beijiang River is the second largest tributary of the Pearl River, and it covers a basin of 52 068 km². The study of chemical weathering and $CO_2$ sink of the Beijiang River Basin is a supplement to the study of carbon cycle of the Pearl River which is the second largest river in China in terms of discharge volume.

**Question 8**: In addition, I have no idea why the authors compared total chemical weathering rate with latitude.

**Answer 8:** Thank you very much for your question. Based on the work of (Meybeck et al., 2006) and other researchers, the chemical weathering rate shows significant spatial trend. Generally it is found that the riverine output of materials is large in the low latitude area due to large runoffs (Fig.C2). So in this study, we compared total chemical weathering rate with latitude to give further evidence to support the conclusion.

[Figure]

Fig.C2 Relative runoff for COSCATs related to mean annual runoff for the exorheic realm (Meybeck et al., 2006)

**Question 9**: Furthermore, there are also some previous studies about the Pearl River and its tributaries, some of which have already taken into consideration anthropogenic weathering in some way. Do the author's HCO3−-basis calculation methods and their results make a difference?

**Answer 9:** Thank you very much for your question. Based on our calculation method, the results in this study have compared with other Chinese rivers, as well as the Xijiang River which is the largest tributary of the Pearl River (see Lines 486-492 in Section 5.2.2). The total of $CO_2$ consumption rates CCR was $823.41\times10^3$ mol $km^{-2}$ $a^{-1}$ in the Beijiang River and was $960\times10^3$ mol $km^{-2}$ $a^{-1}$ in the Xijiang River. The total of $CO_2$ consumption rates in our study area showed little

lower than that in the Xijiang River of the previous study.

In addition, some previous studies calculated the DIC apportionment based on the carbon isotope of DIC, however, our study calculated the DIC apportionment based on mass balance and $HCO_3^-$ concentration, the difference of these two methods will discuss in our other paper. Actually, this manuscript is focused on (1) the chemical weathering rate and the controlling factors on chemical weathering processes, and (2) the temporary sink of $CO_2$ and the influence of sulfide oxidation on net sink of CO2 by DIC apportionment procedure. Thank you very much for your attention to our studies, we hope our study can provide further information for global carbon cycle studies.

**Question 10**: I think the last section in discussion is too descriptive. I also have a concern that temporary and net sink of CO2 show large spatial variations, but in the discussion, the authors mentioned these values only in the SJs station (lowermost part).

**Answer 10:** Thank you very much for your suggestion. Actually, SJs station is the lowest station of the Beijiang River, which can represent the temporary and net sink of $CO_2$ of the whole river basin. In addition, the $CO_2$ net sink of each sub basin were also different and show large spatial variations due to heterogeneity of geology and human activities. The geology showed weak correlation with the $CO_2$ net sink (Fig. 1a), while the $SO_4^{2-}$ have negative correlation with the $CO_2$ net sink (Fig. 1b). It proved that human activities (sulfur acid deposition and AMD) dramatically decreased the $CO_2$ net sink and even make chemical weathering a $CO_2$ source to the atmosphere. We have added this part in the lines 521-524.

[Figure]

**Fig. 1 Correlations between CO₂ net sinks and proportions of proportions of carbonate (a) and correlations between CO₂ net sinks and SO₄²⁻ (b)**

**Question 11**: As shown in equation (21), silicate weathering by sulfuric acid does not affect the concentrations of $HCO_3^-$ in the river. However, in equation (23) and (24), $[SO_4^{2-}]ssw$ seemed to be described as $\alpha CSW \times \alpha SCW / \alpha CCW \times [HCO_3^-]riv$. Would you please explain this calculation?

**Answer 11:** Thank you very much for your question. Firstly, we are very sorry for that there are two equations numbered (23). We have changed numbers in the revision manuscript. In other to explain clearly for this question, we present some of equations as followed.

$$CCW:(Ca_{2-X}Mg_x)(CO_3)_2 + 2H_2CO_3 \rightarrow (2-x)Ca^{2+} + xMg^{2+} + 4HCO_3^- \qquad (18)$$

$$SCW:(Ca_{2-X}Mg_x)(CO_3)_2 + H_2SO_4 \rightarrow (2-x)Ca^{2+} + xMg^{2+} + 2HCO_3^- + SO_4^{2-} \quad (19)$$

$$CSW:CaSiO_3 + 2H_2CO_3 + H_2O \rightarrow Ca^{2+} + H_4SiO_4 + 2HCO_3^- \qquad (20)$$

$$SSW:CaSiO_3 + H_2SO_4 + H_2O \rightarrow Ca^{2+} + H_4SiO_4 + SO_4^{2-} \qquad (21)$$

Where $CaSiO_3$ represents an arbitrary silicate.

(1) If we do not use the equation (22) and (23) as followed, just two equations (24) and (26) can get based on mass balance, however, we have three unknowns ($\alpha_{CCW}$, $\alpha_{SCW}$ and $\alpha_{CSW}$). Thus, we have a hypothesis, according to the studies of (Galy and France-Lanord, 1999) and (Spence

and Telmer, 2005), carbonate and silicate weathering by carbonic acid in the same ratio as carbonate and silicate weathering by sulfuric acid, the mass balance equations are followed:

$$[SO_4^{2-}]_{riv} - [SO_4^{2-}]_{pre} = [SO_4^{2-}]_{SCW} + [SO_4^{2-}]_{SSW} \tag{22}$$

$$[SO_4^{2-}]_{riv} - [SO_4^{2-}]_{pre} = \alpha_{SCW} \times [HCO_3^-]_{riv} \times 0.5 + \frac{\alpha_{csw} \times \alpha_{scw}}{\alpha_{ccw}} \times [HCO_3^-]_{riv} \tag{23}$$

Where the subscripts CCW, SCW, CSW and SSW denotes the four end-members defined by carbonate weathering by carbonic acid, carbonate weathering by sulfuric acid, silicate weathering by carbonic acid and silicate weathering by sulfuric acid, respectively. The parameter α denotes the proportion of DIC derived from each end-member processes.

(2) According to the above equations (22) and (23), we can get a further equation (25) as followed.

$$[Ca^{2+}]_{car} + [Mg^{2+}]_{car} = \alpha_{CCW} \times [HCO_3^-]_{riv} \times 0.5 + \alpha_{SCW} \times [HCO_3^-]_{riv} \tag{24}$$

$$[SO_4^{2-}]_{SCW} + [SO_4^{2-}]_{SSW} = \alpha_{SCW} \times [HCO_3^-]_{riv} \times 0.5 + \frac{\alpha_{csw} \times \alpha_{scw}}{\alpha_{ccw}} \times [HCO_3^-]_{riv} \tag{25}$$

$$\alpha_{CCW} + \alpha_{SCW} + \alpha_{CSW} = 1 \tag{26}$$

(3) Combing the equations (24), (25) and (26), the proportions of HCO$_3^-$ derived from three end-members (CCW, SCW and CSW) can be calculated, and the DIC (equivalent to HCO$_3^-$) fluxes by different chemical weathering processes are calculated by following equations.

$$DIC_{CCW} = \alpha_{CCW} \times [HCO_3^-]_{riv} \tag{27}$$

$$DIC_{SCW} = \alpha_{SCW} \times [HCO_3^-]_{riv} \tag{28}$$

$$DIC_{CSW} = \alpha_{CSW} \times [HCO_3^-]_{riv} \tag{29}$$

**Reference:**

Appelo, C.A.J., Postma, D., 2004. Geochemistry, groundwater and pollution. CRC press.

Meybeck, M., Dürr, H.H., Vörösmarty, C.J., 2006. Global coastal segmentation and its river

catchment contributors: A new look at land‑ocean linkage. Global Biogeochemical Cycles, 20(1).

---

## Author Response (AR2)

**Reply to reviewer comments on "Temporary and net sinks of atmospheric $CO_2$ due to chemical weathering in subtropical catchment with mixing carbonate and silicate lithology" (bg-2019-310)**

Dear Editor and Reviewers,

Thank you very much for your letter and for the reviewers' comments concerning our manuscript entitled "Temporary and net sinks of atmospheric $CO_2$ due to chemical weathering in subtropical catchment with mixing carbonate and silicate lithology" (bg-2019-310). Those comments are all valuable and very helpful for revising and improving our manuscript, as well as the important guiding significance to our researches. We have checked the manuscript and revised it according to the comments very carefully. All the changes have been indicated in an annotated version of the revised manuscript (submission item "Revision, changes marked"). The item-by-item responses to the reviewer comments are followed. Thank you very much for all your help, and we are looking forward to hearing from you.

**Please find the following response to the comments of referees:**

**Responses to Reviewer #1:**

**Question 1**: * alkalinity, DIC and [$HCO_3^-$]

I understood the calculation method of alkalinity, DIC and [$HCO_3^-$] and that these values are similar at pH 7.5–8.5 in this river. However, this explanation is only shown in the response letter. I think concurrent use of these three parameters without special meanings or explanations in the manuscript

makes readers confused. For example, equation (26) can be simply described as $[HCO_3^-]_{CCW}=\alpha_{CCW}\times[HCO_3^-]_{riv}$.

**Answer 1:** Thank you very much for your suggestion. The explanation of DIC and $[HCO_3^-]$ has been added in the revised manuscript (Lines 189-192). The equations (27), (28) and (29) have been corrected as followed, and the corresponding modifications can be seen in Lines 220-222.

$$[HCO_3^-]_{CCW} = \alpha_{CCW} \times [HCO_3^-]_{riv} \tag{27}$$

$$[HCO_3^-]_{SCW} = \alpha_{SCW} \times [HCO_3^-]_{riv} \tag{28}$$

$$[HCO_3^-]_{CSW} = \alpha_{CSW} \times [HCO_3^-]_{riv} \tag{29}$$

**Question 2**: * analytical errors

Thank you for showing the analytical errors of each ion (5%). I think these errors can propagate to the following calculations such as CCW or CCR. In the current manuscript, significant digits of these parameters (as well as ions in Table 1) seem to be too large.

**Answer 2:** Thank you very much for your suggestion. The analytical error of 5% for each ions is an acceptable criterion for major ion analysis. In our lab, the analytical errors are less than 5% and is well controlled within 1%. So the uncertainties of chemical weathering rates introduced by propagation of error is small in the study. In addition, few studies discussed the uncertainties of chemical weathering rates. For example, in the work by (Gaillardet et al., 1999), no discussion on uncertainties. Also in recent works such as (Zeng et al., 2017), (Li et al., 2019) (Wang et al., 2016) and so on.

The significant digits of parameters in Table 1, Table 3 and other part of this manuscript have been reduced. The modified parts have been labelled by red in the manuscript.

**Question 3**: L. 35-37

In the first, second, third, and fourth paragraph in the introduction section, the authors describes the mechanisms of silicate/carbonate weathering, human impact on the chemical weathering, problems in the previous studies in various global rivers, and study area and objectives of this study, respectively. The lines 35-37 are about areas, and should be noted in the third or fourth paragraph.

**Answer 3:** Thank you very much for your suggestion. The sentence "About half of the global CO2 sequestration due to chemical weathering occurs in warm and high runoff regions (Ludwig et al., 1998), so called the hyperactive regions and hotspots (Meybeck et al., 2006)." has been moved to the front of fourth paragraph. The corresponding modifications can be seen in Lines 75-77.

**Question 4**: L. 139-140

The chemical compositions of the rain water should be noted in the result section.

**Answer 4:** Thank you very much for your suggestion. The chemical compositions of the rain water have been added in the result section. The corresponding modifications can be seen in Lines 307-311.

**Question 5**: L. 141

"expert for" reads as "except for".

**Answer 5:** Thank you very much. The "expert for" has been corrected as "except for". The corresponding modifications can be seen in Line 140.

**Question 6**: L. 151

"Fig. 11" reads as "Fig. 2". This figure about stoichiometric analysis is the second figure referred in the manuscript. Please also correct numbers of the following figures.

**Answer 6:** Thank you very much. "Fig. 11" has been corrected as "Fig. 2". The numbers of following figures have also been corrected. The corresponding modifications can be seen in Lines 153-154.

**Question 7**: L. 202/208

There are still two equations numbered as (23). Please correct numbers of the following equations.

**Answer 7:** Thank you very much. The numbers of equations have been corrected.

**Question 8**: L. 517-520

I think these lines and Fig. 12 added in this revision are very important to suggest human impact on the chemical weathering and net sink of $CO_2$. However, the relationship between $[SO_4^{2-}]_{riv}$ and net sink of $CO_2$ seems to be statistically insignificant. Furthermore, I have no idea why the authors use $[SO_4^{2-}]_{riv}$ in this figure. To check the human impact, I think $[SO_4^{2-}]_{scw}$ and $[SO_4^{2-}]_{ssw}$ should be compared.

**Answer 8:** Thank you very much. The correlations between $CO_2$ net sinks and $[SO_4^{2-}]_{SCW}$ or $[SO_4^{2-}]_{SSW}$ have been added in Fig, 12. The corresponding modifications can be seen in Line 523-527.

References:

Gaillardet, J., Dupré, B., Louvat, P., Allègre, C.J., 1999. Global silicate weathering and CO2 consumption rates deduced from the chemistry of large rivers. Chemical Geology, 159(1–4): 3-30.

Li, X. et al., 2019. Hydrochemistry and Dissolved Inorganic Carbon (DIC) Cycling in a Tropical Agricultural River, Mun River Basin, Northeast Thailand. International journal of environmental research and public health, 16(18): 3410.

Wang, L., Zhang, L., Cai, W.-J., Wang, B., Yu, Z., 2016. Consumption of atmospheric CO2 via chemical weathering in the Yellow River basin: The Qinghai–Tibet Plateau is the main contributor to the high dissolved inorganic carbon in the Yellow River. Chemical Geology, 430: 34-44.

Zeng, Q. et al., 2017. Carbonate weathering-related carbon sink fluxes under different land uses: A case study from the Shawan Simulation Test Site, Puding, Southwest China. Chemical Geology, 474: 58-71.

---

## Author Response (AR3)

**Reply to reviewer comments on "Temporary and net sinks of atmospheric CO$_2$ due to chemical weathering in subtropical catchment with mixing carbonate and silicate lithology" (bg-2019-310)**

Dear Editor and Reviewers,

Thank you very much for your letter and for the reviewers' comments concerning our manuscript entitled "Temporary and net sinks of atmospheric CO$_2$ due to chemical weathering in subtropical catchment with mixing carbonate and silicate lithology" (bg-2019-310). Those comments are all valuable and very helpful for revising and improving our manuscript, as well as the important guiding significance to our researches. We have checked the manuscript and revised it according to the comments very carefully. All the changes have been indicated in an annotated version of the revised manuscript (submission item "Revised manuscript "). The item-by-item responses to the reviewer comments are followed. Thank you very much for all your help, and we are looking forward to hearing from you.

**Please find the following response to the comments of referees:**

**Responses to handling editor:**

**Handling editor 's comments:** I think that the manuscript has substantially been improved. The reviewer has evaluated it as reached to the acceptance level. Please make a final step for some technical corrections that the reviewer suggested. You are almost there.

**Response:** Thank you very much for your time and the handling of our manuscript. We greatly appreciate both your help and that of the referees concerning improvement to our manuscript. The technical corrections that the reviewer suggested have been corrected, the revisions have been marked in the revised manuscript and the responses to reviewers are as follows.

**Responses to Reviewer #1:**

**Question 1**: L. 71 Please define "DIC" by spelling it out, as "DIC (dissolved inorganic carbon)".

**Answer 1:** Thank you very much for your suggestion. The "DIC" has been defined as dissolved inorganic carbon. The corresponding modifications can be seen in Line 71.

**Question 2**: L. 138-139 "The hydrochemical compositions of rain water were summarized in Table S1 in the supplementary materials." Please delete this sentence. The rain water data should be noted in the result section, NOT in the material and methods. Furthermore, similar arguments were already shown in the result section (L. 307-311).

**Answer 2:** Thank you very much for your suggestion. The sentence "The hydrochemical compositions of rain water were summarized in Table S1 in the supplementary materials." has been deleted.

**Question 3**: L. 484 There are garbled characters (or unreferred citation in Chinese?).

**Answer 3:** Thank you very much for your reminding. The garbled characters are "Fig.2". The corresponding modifications can be seen in Line 483.

[revised manuscript text omitted]
 | 12.6±9.2 | 12.5±16.3 | 22.9±13.8 | 15.4±18.1 | 25.4±16.0 | 79.0±79.8 | 156.7±206.4 |